



# Impacts of air fraction increase on Arctic sea-ice thickness retrieval during melt season

Evgenii Salganik[1], Odile Crabeck[2,3], Niels Fuchs[4], Nils Hutter[6,5], Philipp Anhaus[5], and Jack Christopher Landy[7]

[1]Norwegian Polar Institute, Fram Centre, Tromsø, Norway
[2]Chemical Oceanography Unit, Université de Liège, Liège, Belgium
[3]Laboratoire de Glaciologie, Université Libre de Bruxelles, Brussels, Belgium
[4]Center for Earth System Sustainability, Institute of Oceanography, Universität Hamburg, Hamburg, Germany
[5]Alfred-Wegener-Institut, Helmholtz-Zentrum für Polar- und Meeresforschung, Bremerhaven, Germany
[6]GEOMAR Helmholtz Centre for Ocean Research, Kiel, Germany
[7]UiT Arctic University of Norway, Tromsø, Norway

**Correspondence:** Evgenii Salganik (salganikea@gmail.com)

**Abstract.** Uncertainties in sea-ice density lead to high uncertainties in ice thickness retrieval from its freeboard. During the MOSAiC expedition, we observed the first-year ice (FYI) freeboard increase by 0.02 m while its thickness decreased by 0.5 m during the Arctic melt season in June–July 2020. Over the same period, the FYI density decreased from 910 kg m$^{-3}$ to 880 kg m$^{-3}$, and the sea-ice air fraction increased from 1 % to 6 %, due to air voids expansion controlled by internal melt. This

increase in air volume substantially affected FYI density and freeboard. Due to differences in sea-ice thermodynamic state (such as salinity and temperature), the air volume expansion is less pronounced in second-year ice (SYI) and has a smaller impact on the density evolution of SYI and ridges. We validated our discrete measurements of FYI density from coring using co-located ice topography from underwater sonar and an airborne laser scanner. Despite decreasing ice thickness, a similar counter-intuitive increasing ice freeboard was observed for the entire 0.9 km$^2$ MOSAiC ice floe, with a stronger freeboard

increase for FYI than for less saline SYI. The surrounding 50 km$^2$ area experienced a slightly lower 0.01 m ice freeboard increase in July 2020, despite comparable half-meter melt rates obtained from ice mass balance buoys. The decreasing draft to thickness ratio from 0.92 to 0.87 observed FYI complicates the retrieval of ice thickness from satellite altimeters during the summer melt season and underlines the importance of considering density changes in retrieval algorithms.

## 1 Introduction

The ultimate goal of mass balance observations of Arctic sea ice is to produce reliable ice thickness measurements throughout the year. Unless sea-ice thickness is directly measured using ice coring, ice mass balance buoys, or ground-based electromagnetic sounding, it relies on remote methods such as airborne or satellite laser (ICESat) and radar (Sentinel-3, CryoSat-2) altimeters (Paul et al., 2018), airborne electromagnetic sounding (Haas et al., 1997), moored upward-looking sonars (Sumata et al., 2023), and underwater sonars mounted on submarines (Lyon, 1961), autonomous underwater vehicles (AUV), or re-

motely operated vehicles (ROV). Except for electromagnetic sounding, these remote techniques estimate either the snow or





sea-ice freeboard (height above the waterline) or the ice draft (height below the waterline). To convert draft or freeboard to sea-ice thickness, they require information on the snow depth and density of both snow and sea ice. Generally, draft measurement from upward-looking sonar reconstructs sea-ice thickness more accurately than freeboard deduced by satellite altimeters since a larger portion of the thickness is measured directly (Mahoney et al., 2015). While satellite altimeters allow for basin-wide studies by measuring the freeboard of snow and sea ice along profiles of the satellite's orbit across the Arctic Ocean, ice mass balance buoys, moored upward-looking sonars, and electromagnetic sounding all require local ground, aerial, or underwater surveys with limited spatial and temporal coverage (Haas et al., 1997; Webster et al., 2022; von Albedyll et al., 2022).

Sea-ice and snow density, as well as snow thickness, play an important role in aerial and satellite altimetry (Dawson et al., 2022), but they are rarely, if ever, measured directly from remote platforms. While the snow and ice densities estimated from in situ historical measurements are assumed to be constant for certain ice types—including first-year ice (FYI), second-year ice (SYI), and multiyear ice (MYI)—snow depth relies on observational climatology, models, or passive microwave observations (Zhou et al., 2021). Inaccuracies in the assumed snow and sea-ice density can lead to substantial errors in the estimated ice thickness (Alexandrov et al., 2010; Landy et al., 2020). For example, typical ice density variations of 882–917 kg m$^{-3}$ can result in variations in sea-ice thickness estimates up to 0.7 m (Kern et al., 2015). The most commonly used sea-ice density data sets for remote sensing purposes, e.g., Alexandrov et al. (2010), have several limitations, including temporal restrictions that prevent them from representing the seasonality of sea-ice density in spaceborne ice thickness products. If sea ice with a typical thickness of 1.5 m (Sumata et al., 2023) has its initial density of 920 kg m$^{-3}$ decreasing to 875 kg m$^{-3}$ during the melting season (Worby et al., 2008), its hydrostatic balance is altered, and the ratio of the freeboard to thickness is expected to decline. This implies that if the ice density transition occurs at a faster rate than the thickness change, the first 0.5 m of its thickness reduction is not detectable in freeboard measurements, as the change in density will counteract the impact of ice melt on the freeboard. Recent advancements have led to the development of new experimental sea-ice freeboard and thickness products from satellite radar altimetry during the Arctic melt season (Dawson et al., 2022; Landy et al., 2022). Landy et al. (2020) estimated potential systematic bias in CryoSat-2-derived sea-ice thickness from various sources of uncertainty, including 16 % and 12 % bias from ice density for FYI and MYI, respectively. Currently, remote sensing ice thickness products often use constant ice density because (1) bulk sea-ice density measurements require challenging in situ discrete sampling, and (2) past studies hypothesise that bulk ice density has a minor impact on ice freeboard and thickness retrieval. However, Kern et al. (2015) noted that sea-ice density is as important as snow depth for using radar altimetry to retrieve sea-ice thickness. Therefore, there is a need to extend accurate sea-ice density data to the entire year so that the seasonal evolution can be realistically accounted for in remotely-sensed ice thickness products.

There are several direct and indirect methods to estimate sea-ice density (Hutchings et al., 2015). It can be directly measured by hydrostatic weighing with 0.1–1.3 % error (Nakawo, 1983) or by the mass/volume method with 3–8 % error (Hutchings et al., 2015), which consists of sizing and weighing extracted ice core sections. However, these measurements are rarely performed at the ice in situ temperatures, as this is not practical and subject to errors from the non-stationary ice temperature and, particularly during the melt season, brine losses from warm sea-ice sections. Assuming sea ice is in hydrostatic equilibrium,





sea-ice density can be indirectly derived from measurements of snow depth, ice freeboard, and draft. The freeboard and draft can be measured directly using in situ field measurements or various combinations of remote methods mentioned earlier.

The bulk density of Arctic and Antarctic FYI varies seasonally, ranging from above 910 kg $^{-3}$ in winter to below 900 kg m$^{-3}$ in summer (Wang et al., 2020; Fons et al., 2023). Timco and Frederking (1996) estimated the typical range of bulk FYI density to be 840–910 kg m$^{-3}$ with 900–940 kg m$^{-3}$ for the density below the waterline. Alexandrov et al. (2010) used ice freeboard

and thickness measurements to estimate FYI density of 917±36 kg m$^{-3}$ from data collected in the Eurasian Russian Arctic during February–May of 1980–1988. Jutila et al. (2022) combined airborne snow depth, ice freeboard, and ice thickness measurements collected in the western Arctic Ocean in April 2017–2019 to estimate densities of 925±18 kg m$^{-3}$ for FYI, 899±17 kg m$^{-3}$ for SYI, and 902±19 kg m$^{-3}$ for MYI. Based on the field measurements from Worby et al. (2008), Fons et al. (2023) reported the seasonality of Antarctic FYI density, which increases from 900 kg m$^{-3}$ in autumn to 920 kg m$^{-3}$

in winter, and decreases from 915 kg m$^{-3}$ in spring toward 875 kg m$^{-3}$ in summer. Using the mass/volume method, Wang et al. (2020) estimated FYI and MYI densities in August to be 793±74 kg m$^{-3}$ and 810±49 kg m$^{-3}$, respectively. Frantz et al. (2019) documented a strong decrease in sea-ice density down to 600 kg m$^{-3}$ as sea ice in Alaska became rotten and melted completely. Most observations show that bulk FYI density outside of melt season is similar to pure ice density of 917 kg m$^{-3}$, while SYI is significantly lighter. Only Frantz et al. (2019), Wang et al. (2020) and Fons et al. (2023) estimated sea-ice density

during the melt season, revealing a large range of values substantially lower than those from winter observations.

Sea-ice density depends on the volumetric fractions of ice, air, and liquid brine. Changes in the air fraction have a greater impact on sea-ice density than changes in brine or ice volume due to the 10 times larger density difference between air and pure ice compared to that between pure ice and brine. Volumetric fractions of brine and air are usually estimated from measurements of bulk sea-ice salinity, density, and temperature (Cox and Weeks, 1983; Leppäranta and Manninen, 1988). Since the air volume

fraction is typically much smaller than the brine volume fraction, it is often neglected in models predicting salinity evolution and brine and ice volume (Griewank and Notz, 2013). During the ice growth phase, the bulk brine fraction of columnar/congelation sea ice is usually below 5 % (Griewank and Notz, 2013) and the air volume fraction below the waterline is usually less than 2 % (Nakawo, 1983; Crabeck et al., 2016). During the freezing period, the temporal and spatial variability of sea-ice density is quite low due to the small variability of both the bulk air and brine fractions. In winter, desalination is mainly governed by the

convective exchange of cold, dense brine with fresher seawater (gravity drainage). In summer, the vertical percolation of fresh surface meltwater through the permeable ice matrix (flushing) controls ice desalination (Griewank and Notz, 2013). Warming drives significant thermodynamic changes that alter the ice, brine, and air volume fractions. Recent Arctic observations show that bulk FYI salinity during the melt season is usually 1–2, with a bulk temperature around -0.8°C (Wang et al., 2020). Considering gas-free sea ice in winter with a negative temperature gradient from -15°C at the surface to -1.8°C at the bottom,

which warms to an isothermal profile of -0.8°C in summer with a salinity of 2, its bulk density should increase from 921 kg m$^{-3}$ to 929 kg m$^{-3}$ due to the increase of brine volume reaching 12 % (Leppäranta and Manninen, 1988). Meanwhile, if we account for an air volume fraction of 5–15%, the bulk ice density should fall within the range of 780–880 kg m$^{-3}$ following Leppäranta and Manninen (1988). Given that small changes in air volume fraction produce large changes in sea-ice bulk density, it is questionable to neglect the impact of air volume fraction on bulk ice density in ice thickness retrieval products. Recent work





indicates that gas exists in sea ice in the dissolved state in brine and in the gaseous phase as air bubbles (Light et al., 2003; Zhou et al., 2013; Moreau et al., 2014; Crabeck et al., 2016, 2019). The distribution of the main atmospheric gases in sea ice is controlled primarily by physical processes such as (1) the initial gas entrapment at the ice-seawater and ice-atmosphere interfaces during ice formation, (2) the potential phase change from dissolved within brine to gaseous in air bubbles or vice versa, that may take place in the brine medium, and (3) brine and bubble transport within sea ice and across the ice-atmosphere

and ice-seawater interfaces. Crabeck et al. (2019) and Light et al. (2003), using thin sections of sea ice, observed the shrinking of gas inclusions upon cooling and expanding during warming, suggesting that temperature exerts a strong control on the gas volume fraction.

Snow plays a crucial role in both the thermodynamics and hydrostatic balance of sea ice. During freezing, snow load depresses the freeboard, while the snowmelt in spring allows for a rebound. In the Arctic, snowmelt can also lead to the

formation of melt ponds (Webster et al., 2022). The areal fraction, depth, and drainage of surface melt ponds significantly affect the sea-ice hydrostatic balance. Initially, when melt ponds form from melting snow, the meltwater typically cannot percolate through the impermeable sea ice below, so it loads the ice similarly to snow and reduces the freeboard (Landy et al., 2014). However, after the onset of flushing, when the sea-ice column is completely permeable, the melt ponds begin to drain towards sea level, and the freeboard isostatically rebounds, usually reducing the ratio of freeboard to thickness. Additionally,

meltwater drainage can lead to the formation of under-ice meltwater layers, which can substantially increase the temperature of sea ice (Salganik et al., 2023c), affecting its brine and air volume, and can also decrease the water density (Smith et al., 2023).

In this study, we present observations of the temporal evolution of sea-ice density, with a focus on its rapid changes during the melt season. We demonstrate how summer changes in sea ice impact the accuracy of ice thickness retrieval from freeboard measurements. We validate our density measurements using freeboard and draft measurements from ice coring, along with

co-located underwater sonar and airborne laser scanner surveys. We assess the representativeness of the summer freeboard evolution with airborne freeboard measurements and validate sea-ice thickness evolution using ice mass balance buoy data. This comparison is made between 1 km$^2$ MOSAiC Central Observatory ice floe and observations from a surrounding area of approximately 50 km$^2$. Additionally, we analyse how the sea-ice density is controlled by its air volume and the corresponding temperature evolution. We present seasonal sea-ice density measurements for different ice types and examine how the observed

seasonality relates to atmospheric and oceanic environmental forcing.

## 2   Methods

The MOSAiC Central Observatory (CO1), settled on an approximately 3 km by 4 km ice floe, drifted across the central Arctic from 4 October 2019 for a period of 10 months (Nicolaus et al., 2022). It followed the Transpolar Drift until it reached the ice edge in Fram Strait and broke apart on 31 July 2020 (Fig. 1a), with an area of 0.9 km$^2$ remaining during summer (CO2, Fig.

1b). We use observations from 23 FYI and 18 SYI coring events to obtain the temporal evolution of draft, freeboard, density, salinity, and temperature of the main unponded ice types at the FYI and SYI coring sites and several ridges (Fig. 1b). To validate and extrapolate coring measurements of ice draft and freeboard to a larger area, we use ice draft measurements from



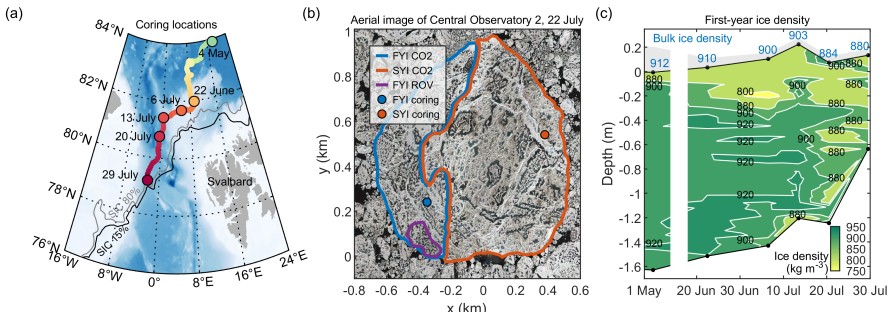

**Figure 1.** Panel **(a)** shows an overview map of the study area with drift of the MOSAiC ice floe for coring observations from 4 May to 29 July 2020. Panel **(b)** shows the locations of first-year ice (FYI) and second-year ice (SYI), coring sites, and ROV sonar surveys on an optical aerial image of Central Observatory (CO2) from 22 July from Neckel et al. (2023). Panel **(c)** shows the contour plot of first-year ice density evolution from hydrostatic weighing with bulk density values for each coring event shown in blue and grey shaded areas representing snow or surface scattering layer thickness. Displayed bathymetric data in **(a)** are from ETOPO2 (National Geophysical Data Center, 2006), and displayed ice edges in **(a)** are derived from the AMSR-2 sea-ice concentration (SIC) product for thresholds of 15 % and 80 % on 29 July 2020 (Spreen et al., 2008).

ROV multibeam sonar (Salganik et al., 2023f) and freeboard measurements from airborne laser scanner (ALS, Hutter et al. (2023b). To validate ice melt rates obtained from ice coring and multibeam sonar, we use observations from ice mass balance
buoys installed at the CO1 and at the Distributed Network (DN), covering a 40 km radius around it (Lei et al., 2022).

## 2.1 Ice coring and ice mass balance buoys

Drilling through the ice, e.g., for ice coring, is the only type of direct measurement that provides both sea-ice freeboard and thickness, while ice mass balance buoys (IMBs) can separately measure ice surface and bottom melt. At each coring site (Fig. 1b), ice cores were extracted using Kovacs Mark II and III Core Barrels; 72.5 mm diameter core barrels were used for
density measurement and 90 mm diameter core barrels for temperature and salinity cores. To minimize spatial variability, FYI and SYI cores were collected within a 130 m distance. However, each core is slightly different and cannot be interpreted as a continuous in-situ time series. One coring event each week yielded a total of 20–30 cores for FYI and SYI sites. After extraction, the freeboard and draft of the cores were measured directly on-site. One temperature profile was made per coring event immediately after the core was extracted, where 4 mm wide holes were drilled into the core with a 0.05 m vertical
resolution. The temperature was measured by inserting the probe of a Testo 720 thermometer into the holes and allowing the temperature to stabilize. After extraction, the density and salinity cores were immediately packed in plastic bags and transported into the ship's freezer room at a temperature of -15°C. Density measurements (Fig. 1c) were performed by sectioning one core per event into 0.04–0.06 m long pieces in a cold lab, followed by hydrostatic density measurements by weighing ice sections in air and kerosene (Pustogvar and Kulyakhtin, 2016). The practical salinity of each melted piece was measured with a YSI 30
conductivity meter. The accuracy of sea-ice salinity measurements significantly affects density measurements due to potential





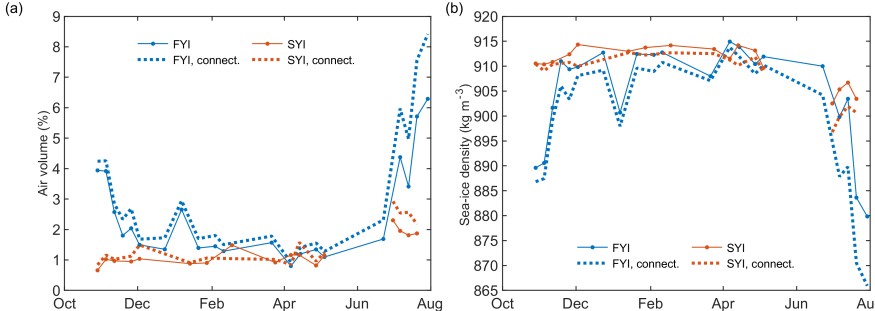

**Figure 2.** Comparison of sea-ice air volume fraction **(a)** and bulk density **(b)** estimated with the two formulations considering disconnected and connected air and brine pockets given in Cox and Weeks (1983).

brine losses. For the hydrostatic method, the error of density measurements from brine loss is limited by 2 % (Pustogvar and Kulyakhtin, 2016).

The relative brine volume of each section was calculated following Cox and Weeks (1983) and Leppäranta and Manninen (1988) for in situ conditions using the ice temperature profile measured in the field. The relative air volume $v'_a$ at laboratory

temperature $T = T_{lab}$ was estimated as follows:

$$v'_a = 1 - \frac{\rho}{\rho_i} + \rho S_i \frac{F_2(T)}{F_1(T)}, \tag{1}$$

where $\rho$ is the sea-ice density measured in the lab, $\rho_i$ is the pure ice density, $S_i$ is the measured sea-ice salinity, $F_1$ and $F_2$ are temperature-dependent coefficients calculated from Cox and Weeks (1983) and Leppäranta and Manninen (1988). This formulation assumes that air bubbles and brine are disconnected and is a conservative (i.e., minimal) estimation of the air

volume fraction, since cooling the ice to laboratory temperature squeezes the air volume due to the expansion of liquid brine to ice during freezing. Reconstructing the air volume fraction at in-situ temperature using the connected formulation in Cox and Weeks (1983) accounts for internal phase changes, particularly the volume of air voids created by the increase in brine volume, which is denser than ice, during warming. This reconstructed air volume fraction at in-situ temperature shows a similar seasonal trend but gives 18±7 % larger FYI air volume estimates compared to laboratory temperature (Fig. 2a). The

connected formulation also gives 5±4 kg m$^{-3}$ lower FYI density values (Fig. 2b). In the following section, we present the conservative estimate of air volume fraction measured directly in the cold room. This is because reconstructing the air volume fraction involves significant error propagation from temperature measurements and brine volume fraction estimates at warmer temperatures, as previously highlighted by Leppäranta and Manninen (1988). Sea-ice density $\rho_{si}$ at in situ temperature $T = T_i$ and disconnected air and brine pockets was calculated as:

$$\rho_{si} = (1 - v'_a) \frac{\rho_i F_1(T)}{F_1(T) - \rho_i S_i F_2(T)} \tag{2}$$

The estimate of sea-ice density from the measurements of ice freeboard and draft, snow thickness and density is referred to here as the effective sea-ice density. Here we used density measurements of snow and surface scattering layer (SSL)—which



is deteriorated granular melting ice similar to large-grained melting snow—from Macfarlane et al. (2023). Given the measured ice thickness $h_i$, ice draft $d_i$, snow/SSL thickness $h_{sn}$, snow/SSL density $\rho_{sn}$, and seawater density $\rho_{sw}$, the effective sea-ice

density can be found as:

$$\rho_{si,eff} = \frac{\rho_{sw}d_i - \rho_{sn}h_{sn}}{h_i} \tag{3}$$

The accumulation of surface meltwater in ponds also impacts the effective sea-ice density by adding extra weight from meltwater, which replaces the lighter sea ice, and by removing the weight of snow or SSL. To estimate sea-ice density while accounting for the impact of surface melt ponds on the hydrostatic balance of an ice floe, we use the near-daily measurements of melt pond

areal fraction and depth from Webster et al. (2022) for CO2, as well as melt pond depth reconstructed from aerial photographs using photogrammetry from Fuchs et al. (2024) for the ROV sonar site (Fig. 1b). The method for incorporating melt ponds and their drainage into sea-ice density estimates is described in Section 2.3.

To study the spatial variability of sea-ice melt, we used non-destructive temperature measurements from IMB buoys. These buoys, which operate with internal heating cycles, allow for precise identification of snow-ice and ice-water interfaces (Jackson

et al., 2013). We employed two types of IMB buoys: 13 SIMBA buoys (Lei et al., 2022) and 5 DTC buoys (Salganik et al., 2023a). The study included direct measurements of sea-ice melt from 18 and 7 IMBs, operating from 22 June to 15 and 28 July, respectively. Two IMBs were located within CO2, ten in CO1, and the remaining at DN. Approximately 20 % of IMBs were installed in level FYI, while the other 80 % were located in level or deformed SYI. We also used IMBs to estimate under-ice water temperature and density outside of coring sites, where water temperature was measured directly.

## 2.2   Airborne Laser Scanner and Remotely Operated underwater Vehicle

Helicopter-based Airborne Laser Scanner (ALS) Riegl VQ-580 provides values of snow, snow-free ice, or melt pond freeboard with kilometer-scale areal coverage, a spatial resolution of 0.5 m, an accuracy of 0.025 m, and an elevation uncertainty of 0.05 m. Freeboard conversion from ALS elevation measurements was performed using an automated open water detection scheme using differences in open water reflectance (Hutter et al., 2023b). In this study, we used nine ALS surveys conducted

on 21 March, 8 and 23 April, 10 May, 30 June, 4, 7, 17, and 22 July 2020. These surveys covered 0.5–0.9 km² of CO2, with the total scanned areas of 75, 49, 57, 58, 49, 5.4, 3.2, 4.7, and 43 km², respectively. To calculate ice freeboard from snow freeboard measured by ALS, we used snow and SSL thickness measurements from near-daily Magnaprobe transects at the CO2 (Webster et al., 2022). In spring, we applied the average transect snow thickness for CO2 and the full ALS scan scale, while for the level FYI site scanned by sonar, we used the average transect snow thickness for level ice as derived by Itkin et al. (2023). ALS

does not provide freeboard measurements of melt ponds, which were located right below the helicopter during surveys and accounted for 3–8 % of the scanned area. The freeboard of these melt ponds was estimated by assuming it was equal to the freeboard of the melt pond edges, using linear interpolation.

Multibeam sonar (DT101, Imagenex, Canada) mounted on a remotely operated vehicle (ROV; M500, Ocean Modules, Sweden; after Katlein et al. (2017)) provides measurements of ice draft over an area approximately 350 m by 200 m (Fig. 1b),

with 0.05 m vertical accuracy and 0.5 m horizontal resolution (Coppolaro, 2018). Surveys were conducted in a grid pattern with



a line spacing of 20–25 m. Seven surveys at a depth of 20 m with nearly 100 % overlap were performed during the melt season (from 24 June to 28 July 2020), near the floe edge of the CO2 ice floe. In this study, we focus on the FYI part of sonar surveys that was not affected by false bottoms (Salganik et al., 2023c). To account for the time difference of 1–3 days between ALS and ROV surveys in summer, we linearly interpolated ALS freeboard and ROV draft measurements to align them temporally

and compute the bulk sea-ice density.

Kilometer scale ALS surveys include areas with different ice types, and notably ridges. Ridges occupy a substantial fraction of the ice pack (Melling and Riedel, 1996), affecting the mass and hydrostatic balance of sea ice (Salganik et al., 2023e). Based on measured snow freeboard, we classified ALS observations into two classes: non-ridged sea ice, referred to here as level ice, and ridged sea ice, in order to compare ALS freeboard evolution of different ice types and to remove freeboard biases

caused by the spatial variability of the ridge areal fraction. For summer observations, characterized by thin snow (Webster et al., 2022), we identified ridges using a 0.35 m threshold above the modal snow freeboard. The snow freeboard threshold corresponds to the sail width of a ridge, investigated by both ALS and ROV sonar on 30 June. The estimated average ratio of the keel and sail width was 2.7 from the co-located ridge surface and bottom topography, which aligns with the ratio of 3 reported by Strub-Klein and Sudom (2012). For spring observations with substantially thicker snow (Itkin et al., 2023), we

used a 0.5 m threshold above the modal snow freeboard following Ricker et al. (2023).

## 2.3 Melt ponds

We distinguish the surface topography of ice covered with melt ponds (ponded) and not covered with melt ponds (unponded). Ponded areas are initially undrained but may become drained, when meltwater is partially or completely transferred from the melt pond to the ocean (Eicken et al., 2002). Unponded ice located next to drained melt ponds is also referred to here as

drained. The uplift of sea ice following melt pond drainage can be described analytically with the parameterization from Landy et al. (2014). For drained melt ponds, there is a noticeable difference between the freeboard of ponded ice and the unponded ice around it. This difference depends on whether melt ponds are deep enough to retain some meltwater after drainage, which happens when the initial melt pond depth $h_{mp}$ exceeds the freeboard of drained unponded ice surrounding the drained melt pond. For shallow melt ponds with zero melt pond depth after drainage, the freeboard of drained unponded ice $fb_{d,up}$ can be

found as:

$$fb_{d,up} = h_i - \frac{\rho_{si}(h_i - a_{mp}h_{mp}) + \rho_{sn}h_{sn}(1 - a_{mp})}{\rho_{sw}}, \tag{4}$$

where $a_{mp}$ is the melt pond fraction. The freeboard of ponded areas after pond drainage is $fb_{d,up} - h_{mp}$. For melt ponds deeper than freeboard of unponded ice, the unponded ice freeboard equals:

$$fb_{d,up,deep} = \frac{(\rho_{sw} - \rho_{si})(h_i - a_{mp}h_{mp}) - \rho_{sn}h_{sn}(1 - a_{mp})}{\rho_{sw}(1 - a_{mp})} \tag{5}$$

The freeboard of sea ice before melt pond drainage can be found as:

$$fb_{ud} = \frac{(\rho_{sw} - \rho_{si})(h_i - a_{mp}h_{mp}) - \rho_{sn}h_{sn}(1 - a_{mp})}{\rho_{sw}} \tag{6}$$



For undrained ponds, we assume that the freeboard is the same for both ponded and unponded areas. For ponded and drained ice with deep ponds, we assume a zero melt pond freeboard. For drained melt ponds, the effective sea-ice density depends on whether the freeboard of ponded areas is considered. For each scenario (drained or undrained) and ice type (ponded or unponded), the effective sea-ice density can be found from the corresponding freeboard value using Eq. (3). If the freeboard of both ponded and unponded areas $fb_i$ is measured, for the meltwater density $\rho_{mw}$, the effective sea-ice density can be found as:

$$\rho_{si,eff} = \frac{\rho_{sw}d_i - \rho_{sn}h_{sn}(1-a_{mp}) - \rho_{mw}h_{mp}a_{mp}}{(d_i + fb_i)(1-a_{mp}) + (d_i + fb_i - h_{mp})a_{mp}} \tag{7}$$

This equation allows to estimate the effective sea-ice density based on the measurements of snow/SSL/melt pond freeboard from ALS, ice draft from sonar, snow thickness from transect, melt pond depth and area from photogrammetry, and snow density from snow pits.

## 3   Results

We present observations of temporal evolution of ice thickness and freeboard, sea-ice density, and air and brine volume during melt season through spring and summer. We validate density measurements from hydrostatic weighing using various measurements of ice freeboard and draft, including ice coring and comparison to co-located ALS and ROV sonar surveys. We further upscale freeboard measurements using ALS surveys, and we validate sea-ice melt measurements using data from IMB buoys. Finally, we present the seasonal evolution of sea-ice physical parameters for undeformed first- and second-year ice and ridges.

### 3.1   Temporal evolution of snow and ice thickness and freeboard during melt season

The melt season was characterized by a rapid decrease in ice thickness, starting in late June. All observing platforms (i.e., FYI coring, ROV sonar, and IMBs) showed an ice draft of around 1.4 m in late June, which fell below 1 m by the end of July (**Fig. 3e**). Level ice thickness decreased by 0.51 m from 22 June to 29 July at the FYI coring site, by 0.56 m at the FYI ROV sonar site, and by 0.67±0.20 m at IMBs at DN in a 40 km radius around the main ice floe (Fig. 3d). Thicker ice types, such as SYI and ridges, experienced more rapid melt than FYI, with total ice melt measured as 0.93 m and 1.77 m, respectively, by ROV sonar between 24 June and 28 July (**Fig. 3d**). Additionally, FYI coring site was affected by the presence of false bottoms, which led to reduced ice melt (Salganik et al., 2023c; Raphael et al., 2024). While all observing platforms agreed on the magnitude and timing of the melt, there were minor differences in the recorded values.

Despite the decrease in ice thickness during late June and July, the ice freeboard increased at both FYI coring and FYI ROV sites (**Fig. 3a**), and across the entire CO2 floe (**Fig. 3b**). At the FYI coring site, the ice freeboard was initially 0.13±0.04 m on 22 June and increased to 0.14–0.15 m during 6–29 July. Similarly, ALS measurements at FYI ROV site between 30 June and 22 July showed a 0.02 m ice freeboard increase and a 0.01 m snow freeboard decrease (Fig. 4a–d). The ALS measurements on 4 July, which deviated from other values, were considered uncertain and not used for sea-ice density retrieval. At the FYI coring site, the freeboard increased rapidly to 0.22±0.05 m during a melt pond drainage event on 13 July (Fig. 3a).



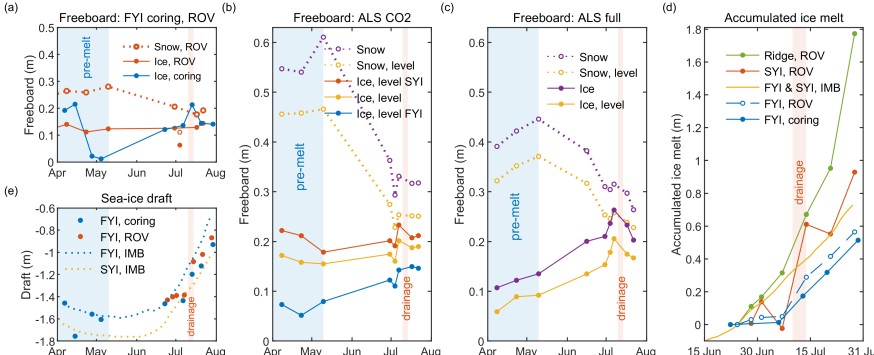

**Figure 3.** Panels **(a)**, **(b)** and **(c)** show freeboard evolution of various ice types for small-scale (FYI coring and ROV sites), floe-scale (CO2) and large-scale (ALS full survey), respectively. Panel **(d)** shows accumulated ice melt measured by coring, ice mass balance buoys (IMB), and ROV underwater sonar for ridge, first-year ice (FYI) and second-year ice (SYI) after 24 June. Panel **(e)** shows ice draft measured by FYI coring, ROV sonar, and estimated from IMB measurements of snow and ice thickness, assuming hydrostatic equilibrium.

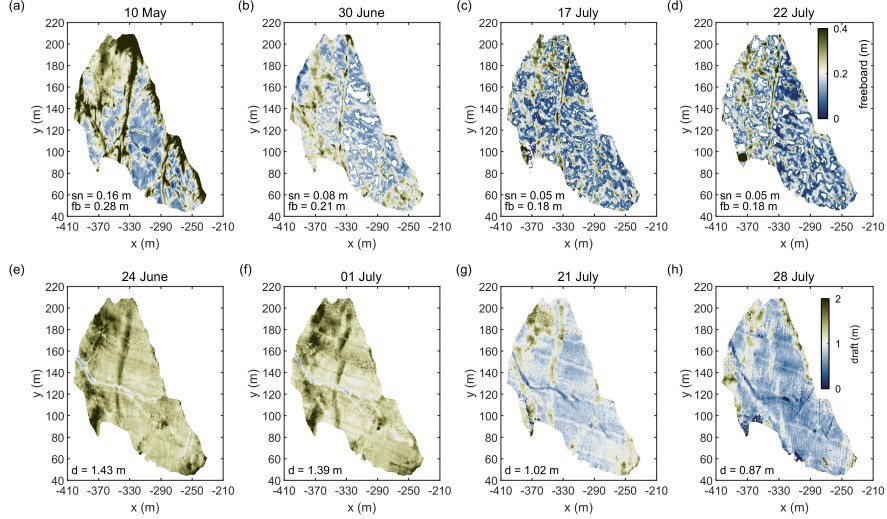

**Figure 4.** Snow or surface scattering layer freeboard from airborne laser scanner **(a–d)** and ice draft from underwater sonar surveys **(e–h)** for selected dates covering the first-year ice area of sonar surveys marked in purple on Fig. 1b. Contour plot colours follow recommendations of scientifically derived colour maps (Crameri et al., 2020).

For the CO2 ice floe from June to July, the areal fractions of level FYI, level SYI, and ridge sails were 31 %, 57 %, and 12 %, respectively. On May 10, the pre-melt ice freeboard measured by ALS was 0.12±0.11 m for level FYI and 0.20±0.15 m for level SYI (Fig. 3b). The pre-melt level ice freeboard on 10 May for CO2 and the full ALS scan were 0.17±0.14 m and 0.15±0.12 m, respectively (Fig. 3b,c). From ALS surveys on 30 June and 22 July, the CO2 ice freeboard increased by 0.029 m, with a 0.036 m increase for level FYI and no change for level SYI. For the full ALS coverage, the ice freeboard of both





ridges and level ice decreased by 0.008 m, while the freeboard of level ice increased by 0.014 m (Fig. 3c). Sea-ice freeboard evolution from May to July was consistent across different scales from FYI coring and ROV sonar sites (0.014 km$^2$) to ALS

CO2 (0.9 km$^2$) and ALS full coverage (40–50 km$^2$), with overall minor changes in ice freeboard during July (Fig. 3a–c).

    The pre-melt snow thickness in April–May was 0.16–0.18 m for coring and snow pit observations at FYI coring site, while the bulk pre-melt snow density was 300±3 kg m$^{-3}$ at FYI coring site and 293±48 kg m$^{-3}$ for the whole CO1. Snow began to melt on 10 May, leaving only SSL on level ice by 4 July, according to IMB observations. Snow or SSL thickness at FYI coring site linearly decreased from 0.08–0.10 m on 22 June to 0.07 m on 4 July. On 4–29 July, SSL thickness was 0.05–0.07 m at both

FYI coring site and the level ice part of the transect (Webster et al. (2022), Fig. 3a). During the melt season in June–July, the bulk snow or SSL density was 390±60 kg m$^{-3}$ at coring sites (Macfarlane et al., 2023).

### 3.2   Temporal evolution of sea-ice density during melt season

A strong decrease in bulk FYI density was recorded by all observing platforms during the summer melt. Discrete measurements of bulk sea-ice density based on hydrostatic weighing at the FYI coring site gradually decreased from 910 kg m$^{-3}$ on 22 June

to 880 kg m$^{-3}$ on 29 July, with a pre-melt density of 912±2 kg m$^{-3}$ during 15 coring events from 14 November to 4 May (Fig. 5b). The effective density estimates from the measured freeboard and draft at FYI coring site showed a similar decrease in summer, from 909 kg m$^{-3}$ on 22 June to 860 kg m$^{-3}$ on 29 July, with a pre-melt density of 918±33 kg m$^{-3}$ and a drop in bulk ice density during a melt pond drainage event. A combination of ALS freeboard and ROV sonar draft measurements produces a mean effective FYI density with a pre-melt value of 926±4 kg m$^{-3}$ in March–May, which decreased to 912 kg m$^{-3}$ on 30 June

and further to 890 kg m$^{-3}$ on 17 July and 884 kg m$^{-3}$ on 22 July (Fig. 5b). The pre-melt estimates of density are calculated using four freeboard measurements from ALS scans in March–May (Fig. 5a) and ice draft measured by ROV on 30 June assuming ice growth and melt in March–May from IMB level ice measurements, with a thickness difference of 0.07–0.24 m from 30 June FYI thickness of 1.52 m (Fig. 5d). The effective ice density, derived from Eq. (7) to account for melt ponds at FYI ROV site, was computed using an average melt pond depth of 0.07–0.10 m and an areal coverage of 0.21–0.24. The effective

ice density considering melt pond extent decreased from 906 kg m$^{-3}$ on 22 June to 882 kg m$^{-3}$ and 876 kg m$^{-3}$ on 17 and 22 July, respectively.

### 3.3   Temporal evolution of air and brine volume during melt season

The estimate of FYI brine volume, obtained using coring measurements of ice salinity and temperature, increased from 6±1 % in December–June (15 cores) to 19±4 % in July (4 cores). Both air volume fractions derived from discrete measurements

of ice density using hydrostatic weighing and effective density from freeboard measurements using Eq. (1) showed a strong increase from May to July (Fig. 5c). FYI air fraction, obtained using hydrostatic weighing, increased from 1.4±0.3 % in December–June (15 cores) to 3.4–6.3 % in July (4 cores). FYI air volume had, on average, 2.5 times larger effect on bulk FYI density than brine volume. While discrete measurements from hydrostatic weighing density showed a slight decrease during the melt pond drainage event, the air volume fraction from in-situ freeboard density showed an increase. SYI air volume doubled,

from 1.0±0.2 % in the cold season (October to May) to 2.0±0.2 % in July.





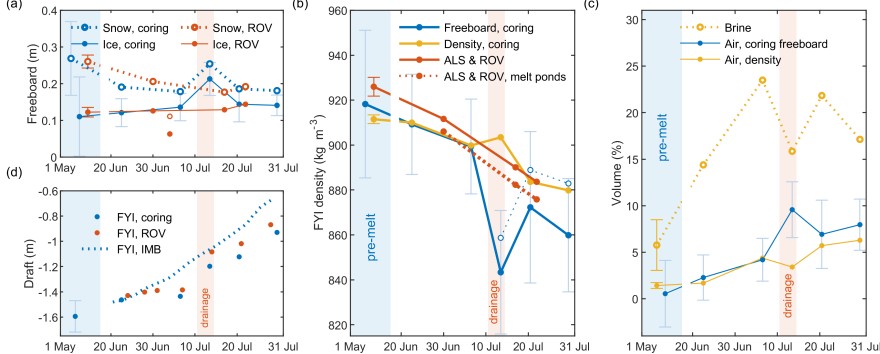

**Figure 5.** Summer evolution of first-year ice (FYI) snow and ice freeboard **(a)**, bulk density **(b)**, brine and air volume **(c)** and draft **(d)** from density and freeboard measurements from FYI coring and a combination of co-located airborne-laser scanner (ALS) freeboard and ROV sonar draft measurements at FYI ROV site. Density measurements (yellow lines) were made using hydrostatic weighing, while freeboard measurements (blue lines) at the coring site were made using thickness tape. Error bars represent one standard deviation of weekly measurements in summer or of all pre-melt weekly measurements. Dotted blue line in **(b)** represents sea-ice density estimate corrected for coring freeboard measurements performed in unponded areas as described in Section 4.4.

Previous studies have argued that ice coring can lead to substantial brine loss during the ice core extraction, particularly in permeable sea ice, which can produce an artificial increase in air volume fraction. Here, the air volume fraction computed from the effective ice density using in situ freeboard measurements shows a similar trend to the air volume fraction computed from bulk ice density using the hydrostatic weighing after ice core extraction. The agreement between the air volume estimates from

in situ freeboard measurement and from the extracted ice cores confirms that the increase of air volume during the melt period is not linked to brine loss during ice core extraction.

### 3.4 Seasonal evolution of sea-ice physical parameters

The freezing season was characterized by a strong decrease in bulk sea-ice temperature, from -2°C in October to -12°C in March (Fig. 6d). As the FYI thickened from 0.4 m in October to 2.0 m in March, the ice freeboard increased from 0.03 m to

0.24 m, while the snow layer depth above FYI was quite steady, ranging from 0.08 m to 0.12 m (Fig. 6a,c,e). In spring, the snow layer became thicker, reaching 0.21 m in early May. Most of this snow layer melted within the following month, reducing to 0.05 m by the end of June. Throughout the winter, the sea ice slowly underwent desalination, with the process accelerating in early June, resulting in a bulk ice salinity close to 1 by the end of July (Fig. 6b). The SYI followed a similar cycle but exhibited lower bulk ice salinity and brine volume fraction (Fig. 6b,f). SYI experienced a gradual increase in freeboard during winter

and spring, with comparable ice and snow thickness, and with unchanging freeboard in June and July (Fig. 6a,c,e).

We observed a strong seasonality in the air volume of FYI and a weaker seasonality for SYI. At the beginning of the growing season (end of October), the mean air volume fraction of FYI was 3.9 %. It quickly decreased to below 2 % by early December as the ice cover cooled and became mostly impermeable, with the mean brine volume fraction dropping below 5 % (Golden



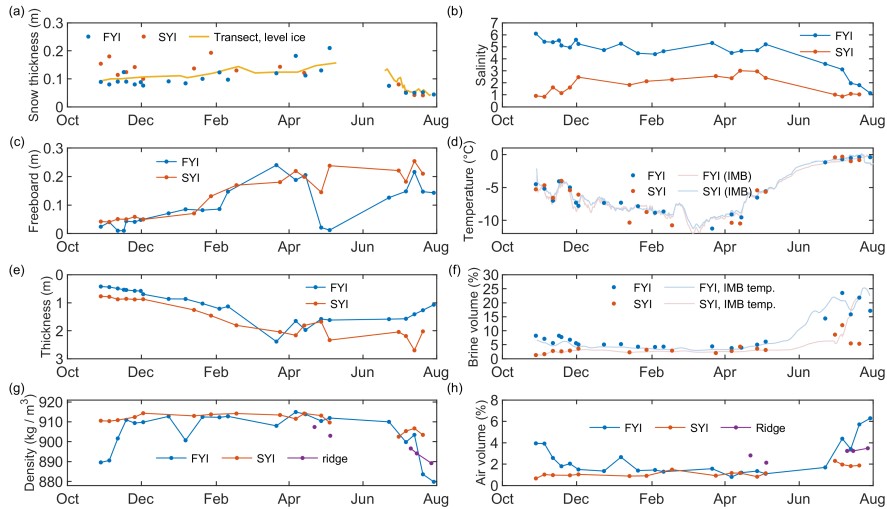

**Figure 6.** Annual evolution of snow thickness **(a)**, bulk sea-ice salinity **(b)**, freeboard **(c)**, temperature **(d)**, thickness **(e)**, brine volume **(f)**, density **(g)**, and air volume **(h)** for level first-year ice (FYI), level second-year ice (SYI) and ridges from the coring observations during MOSAiC in 2019–2020.

et al., 1998). The rapid decrease of FYI air fraction, from 3.9 % on 4 November to 1.8 % on 18 November, occurred within

two weeks, at an average rate of -1.1 % per week (Fig. 6h). During the same period, the bulk FYI density increased from $893\pm10\,\mathrm{kg\,m^{-3}}$ to $908\pm8\,\mathrm{kg\,m^{-3}}$ (Fig. 6g). Throughout the winter, the ice column remained mostly impermeable, except for the warm bottom layer, and the mean air volume stayed below 2 % (Fig. 6f,h). The bulk FYI density remained stable around $912\,\mathrm{kg\,m^{-3}}$. At the end of May, the ice column became isothermal and fully permeable, with a brine volume fraction largely above the 5 % threshold (Fig. 6f). As the ice warmed and the brine volume fraction increased, the mean air volume fraction

tripled in the ice column, reaching a mean value of 6.3 %. Simultaneously, the bulk ice density dropped below $880\,\mathrm{kg\,m^{-3}}$. Ice coring conducted in August–September 2020 at the MOSAiC CO3 (the same geographical location as CO2 in winter, Nicolaus et al. (2022)) with a thickness of $1.7\pm0.3$ m, showed a bulk air volume of $3.7\pm0.3$ % and a bulk sea-ice density of $894\pm4\,\mathrm{kg\,m^{-3}}$ for four unponded and two ponded level ice cores.

  SYI air volume was stable during the freezing period and doubled from $1.0\pm0.2$ % in the cold season (October to May) to

$2.0\pm0.2$ % in the summer in July (Fig. 6h). Consequently, the bulk SYI density decreased from $912\pm2\,\mathrm{kg\,m^{-3}}$ in October–May to $905\pm2\,\mathrm{kg\,m^{-3}}$ in June–July. Unlike FYI, which experienced a gradual temperature increase in July, SYI slightly cooled during this period. The observed decrease in bulk SYI temperature at the coring site by around 1°C in mid-July was not fully representative according to IMB temperature measurements (Fig. 7a). Four different ridges were sampled from 22 April to 5 May and from 10 to 27 July. The air fraction in these ridges was 2.1–2.8 % in April–May and 3.2–3.5 % in July, exhibiting a

similar decreasing density trend to FYI (Fig. 6g).

    The maximum ice thickness of FYI and SYI, as measured by IMBs, was 1.75 m and 1.94 m, respectively (Fig. 3e). This relatively small difference in thickness is because SYI at CO1 was formed in December 2018 and had a mean thickness of



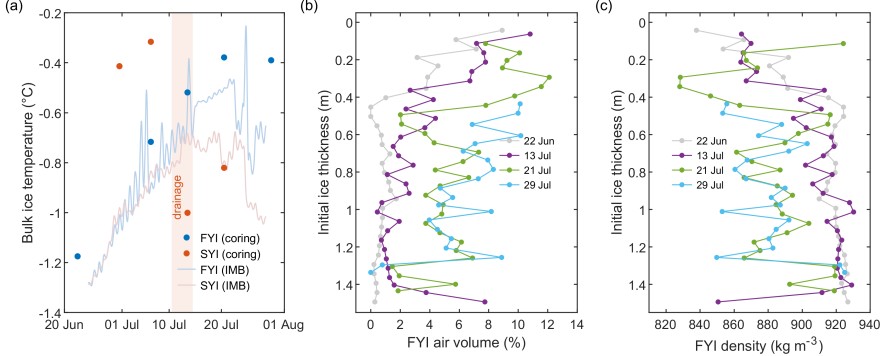

**Figure 7.** First-year ice (FYI) and second-year ice (SYI) bulk temperature from coring sites and ice mass balance buoys (IMBs) **(a)** and vertical profiles of FYI air volume **(b)** and density **(d)** during summer.

0.7–1.0 m by October 2019 (Krumpen et al., 2020). Additionally, the low-salinity residual surface layer at the SYI coring site was typically 0.6–0.8 m thick (Salganik et al., 2023c). This indicates that 60–70 % of the pre-melt SYI thickness was
formed during 2019–2020, sharing similar physical properties with FYI. We focused on FYI physical properties due to the less extensive sampling of SYI. This includes large spatial variability in ice thickness at SYI coring and sonar sites (Fig. 6e) and non-representative SYI temperatures during the melt season (Fig. 7a). The SYI thickness at the coring site ranged from 2.0 to 2.6 m, with no clear decreasing trend during the melt season. Therefore, SYI density estimates from hydrostatic weighing were validated only using ALS ice freeboard estimates at CO2. Observations of increasing SYI freeboard at CO2 suggest a similar
decreasing trend in bulk densities for both SYI and FYI (Fig. 3b). For a constant sea-ice density, the average ice thickness decrease of 0.67 m measured by IMBs would typically result in a 0.07 m decrease in freeboard. However, both FYI and SYI at CO2 experienced similar ice freeboard increases of 0.02 m and 0.01 m, respectively, suggesting similar changes in their bulk densities and air volume fractions.

## 4   Discussion

### 4.1   Comparison of methods for sea-ice density estimation

Our observed seasonal variability of FYI density closely matches the Antarctic FYI density estimates from Fons et al. (2023), derived from ship-based measurements from Worby et al. (2008), with summer and winter density values of 875 kg m$^{-3}$ and 920 kg m$^{-3}$, respectively. The FYI density estimates for February–May by Alexandrov et al. (2010), based on in-situ freeboard and draft measurements, indicated similar values of 917±36 kg m$^{-3}$. The FYI and SYI estimates from April by Jutila et al.
(2022) were 925±18 kg m$^{-3}$ and 899±17 kg m$^{-3}$, respectively, showing better agreement for FYI but larger discrepancies for SYI. Unlike our observations, which showed similar FYI and SYI thickness (Fig. 6b), Jutila et al. (2022) observed SYI to



be 3.2–4.5 m thick, 3–6 times thicker than adjacent FYI. Our airborne pre-melt FYI density estimates of 926±4 kg m$^{-3}$ were higher than our ground-based measurements but identical to the airborne estimates by Jutila et al. (2022) (Fig. 5b).

Nearly all methods of measuring sea-ice density have their disadvantages and limitations. Ideally, hydrostatic weighing
should be performed at in situ temperatures, but such measurements are time-consuming, weather-dependent, and may suffer from larger brine losses when sectioning warm ice. The hydrostatic weighing method used in this study provides the highest accuracy (Fig. 5b), as low laboratory temperatures limit brine loss during ice sectioning, and kerosene coating minimizes brine losses at the interface of core sections by occupying the emptied brine volume (Pustogvar and Kulyakhtin, 2016). Past studies have argued that ice coring leads to substantial brine loss during ice core extraction, particularly in permeable sea ice
(Notz, 2005; Hutchings et al., 2015). This brine loss can artificially increase the air volume fraction and underestimate bulk ice density. In our study, the bulk ice density and the air volume fraction, computed from bulk ice density deduced in situ from ice freeboard and ice thickness measurements, show a similar trend to those computed from the hydrostatic method. However, the average pre-melt bulk density from hydrostatic weighing was 6 kg m$^{-3}$ lower than from direct measurements of snow and ice freeboard and thickness (Fig. 5b), which may be considered assuming 0.6 % brine volume loss or considering ice surface
roughness, described in Section 4.4. During June–July, FYI density was equal or slightly larger than the effective FYI density at coring and ROV sites, indicating no impact of brine loss on hydrostatic weighing density measurements during the melt season. During the same period, the decrease in bulk FYI density was 30 kg m$^{-3}$ from weighing, 26–49 kg m$^{-3}$ based on coring freeboard data, and by 28–30 kg m$^{-3}$ from co-located ALS and ROV observations (Fig. 5b).

Estimates of effective sea-ice density from ice freeboard and thickness are not influenced by brine losses but typically exhibit
much larger spatial variability of 10–30 kg m$^{-3}$ than observations from sea-ice weighing (Alexandrov et al., 2010; Jutila et al., 2022). Winter observations (14 coring events) using the weighing showed a bulk FYI density standard deviation of 2 kg m$^{-3}$ (i.e. 912±2 kg m$^{-3}$) while ice freeboard measurements at the coring site gave a standard deviation of 33 kg m$^{-3}$ (i.e. density of 920±33 kg m$^{-3}$), despite the freeboard being an average value of around 20–30 cores taken weekly, covering an area of around 10 m by 10 m (Fig. 5b). This is likely related to the large spatial variability of ice freeboard and draft (Fig. 6b), rather than the
spatial variability of in-situ sea-ice density, as sea ice cannot bend to account for meter-scale deviations from its hydrostatic balance (Fuchs et al., 2024). Indeed, the standard deviation of bulk FYI and SYI density collected in Nansen Basin in November 2022 using hydrostatic weighing was only 5 kg m$^{-3}$ for 9 FYI and 7 SYI cores located 5 m from each other (Salganik et al., 2023b). This supports the notion of small meter-scale spatial variability in sea-ice density, despite much larger variability in the effective sea-ice density. Bulk sea-ice density depends on sea ice's thermodynamic state (i.e., temperature, salinity), which
determines the volume fraction of ice, brine, and air, and has lower spatial variability than ice thickness and freeboard, which are used to estimate effective ice density. Similarly, ALS observations revealed large meter-scale spatial variability in the SSL freeboard (Fig. 8a,d) despite small spatial variability in the SSL thickness (Webster et al., 2022). We estimated the small-scale spatial distribution of the effective sea-ice density using Eq. (3) smoothing the data from the ALS freeboard and ROV sonar draft over a 10 m window (as for coring). The results showed significant spatial variability, ranging from 15 to 32 kg m$^{-3}$ (Fig.
8). We demonstrated that hydrostatic weighing is one of the most reliable and affordable methods for measuring sea-ice density. Unlike other methods, it does not rely on the assumption that gravity and buoyancy forces acting on sea ice are in hydrostatic



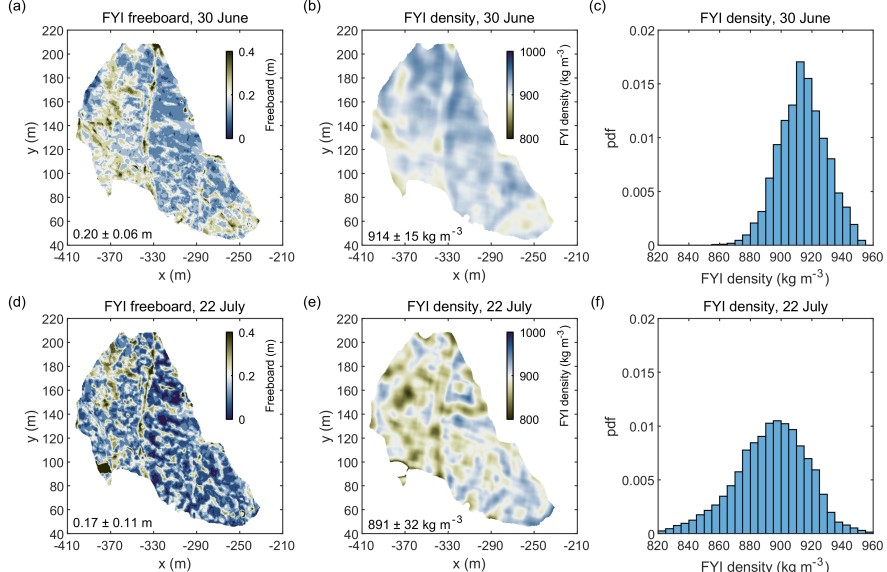

**Figure 8.** Contour plot of snow freeboard from airborne laser scanner **(a, d)**, contour plot of effective sea-ice density **(b, e)** and histogram of effective sea-ice density **(c, f)** for co-located draft from sonar and freeboard from ALS for 30 June and 22 July. The effective sea-ice density was estimated using Eq. (3) from freeboard and draft values, smoothed using two-dimensional convolution with a 10 m wide window.

equilibrium. Our measurements of sea-ice freeboard and draft indicate that this equilibrium is not locally satisfied, resulting in one order of magnitude greater spatial variability in sea-ice density for methods based on this assumption compared to the direct density measurements from hydrostatic weighing. For future studies, it is beneficial to sample sea ice with representative temperature and thickness evolution, which can pose a challenge for SYI and MYI.

### 4.2   Importance of sea-ice density evolution for ice thickness retrieval

The results of the full seasonal evolution of snow and ice thickness and freeboard, and bulk sea-ice density confirmed previous findings (Alexandrov et al., 2010) that during the growth, the ice freeboard is primarily influenced by the increase in ice thickness (Fig. 6c,e). In the spring, when sea ice reaches a steady state, the freeboard is controlled by the snow load (Fig. 6a,c). From May to the end of June, the observed increase in sea-ice freeboard was mainly due to a decrease in snow thickness. IMBs recorded 0.16 m of snowmelt during this period, with no ice melt occurring before 15 June.

While a decrease in freeboard during ice melt is expected, detailed observations during the melt season revealed an increase in sea-ice freeboard despite significant ice melt. Since most of the snow layer melted before early July, this freeboard increase is attributed to a reduction in bulk ice density, which enhances sea-ice buoyancy. This was confirmed by ice freeboard measurements at FYI and SYI coring sites, along with a combination of ALS measurements of snow freeboard and transect measurements of snow and SSL thickness for both level FYI and SYI within CO2. Ice melt was validated using the thickness





measurements at the FYI coring site, sonar draft measurements of FYI and SYI, and IMB measurements of FYI and SYI thickness within DN.

The ice thickness estimates from the CryoSat-2 radar altimeter, with 80-km resolution, indicated a 0.62 m ice melt from
22 June to 23 July around the CO2 location, closely matching the 0.64 m ice melt measured by IMBs for the same period. However, CryoSat-2 also overestimated sea-ice melt by 1.5 m in May–June 2022 and detected an ice melt onset one month earlier than actual (Landy et al., 2022), while IMBs measured only 0.06 m ice melt before 22 June and no melt before 15 June. Additionally, the snow freeboard for ALS full survey decreased by 0.14 m in May–June and by 0.05 m in July. This suggests that CryoSat-2 cannot penetrate through the moist snow and SSL starting from the onset of snow melt. Incorporating
a physics-based parameterization of sea-ice density and SSL thickness could enhance the accuracy of altimetry measurements.

## 4.3 Geochemistry of air volume evolution

The bulk density of sea ice is determined by the volume fractions of its constituent phases: ice, brine, and air. Among these, the air volume has the most significant impact on the bulk density and buoyancy. However, mathematical models of FYI often neglect the air volume fraction, based on the premise that the brine volume fraction is significantly larger than the air volume
fraction (Griewank and Notz, 2013). While this assumption can be valid, treating the air volume fraction as constant and negligible can lead to significant errors in estimating bulk ice density. Air has a very low density (1 kg m$^{-3}$) compared to other media (1000 kg m$^{-3}$ for water and 917 kg m$^{-3}$ for pure ice), so even small variations in the air volume fraction can lead to substantial changes in bulk ice density. As illustrated in Fig. 5b and Fig. 7c, the observed decrease in bulk FYI density during the melt season corresponds to an increase in air volume fraction throughout the ice column. The air volume fraction
in columnar ice exceeded 4 %, reaching over 10 % in the sea-ice surface layer above the waterline (Fig. 9a and Fig. 7). This increase is synchronized with the rise in the brine volume fraction as the ice warms, indicating a strong link between the air volume fraction and the thermodynamic state of sea ice (Fig. 6). The ice above the waterline shows a systematic enrichment in the air volume fraction compared to the ice below the waterline (Fig. 9a,d) and for FYI, this appears decoupled from the thermodynamic state of sea ice, as the air volume is not correlated with the brine volume (Fig. 9b,c). This observation aligns
with previous studies indicating that the air volume fraction in the upper ice layer is controlled by the formation of granular ice, snow ice, and superimposed ice. Frazil ice, which traps gas directly from the atmosphere during its formation, contains more gas than columnar ice (Tsurikov, 1979; Perovich and Gow, 1996; Cole et al., 2004; Zhou et al., 2013; Crabeck et al., 2016). Additionally, the random crystal alignment in frazil ice also reduces the efficiency of drainage processes that expel dissolved gases (Golden et al., 1998). As sea ice grows, the air volume fraction in the surface layers increases due to the formation of
snow ice and superimposed ice, which trap air bubbles inherited from the overlying porous snow cover (Cole et al., 2004; Crabeck et al., 2016). During the melt period, the ice above the freeboard becomes enriched in air as brine is drained and replaced by air. During the melt, as shown in Fig. 9a,b, the increase in air volume fraction is more pronounced in the ice above the freeboard, especially in the heavily desalinized ice layer (small plain red dots in Fig. 9b). This is because the flushing process leaves the previously brine-filled pockets in the ice layer above the waterline filled with air (Perovich and Gow, 1996).





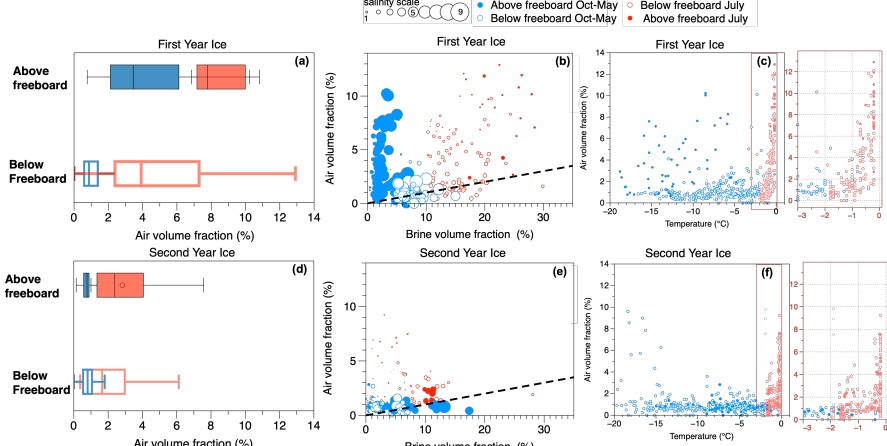

**Figure 9.** First-year ice (FYI) and second-year ice (SYI) air volume distribution **(a, d)**, and the relationship between the air volume fraction and the brine volume fraction **(b, e)** and temperature **(c, f)**. The dashed lines in **(b)** and **(e)** represent the volume of air formed due to a 10 % density difference between pure ice and brine during melt. The diameters of the circles in panels **(b)** and **(e)** are proportional to the ice sample salinity; smaller circles indicate lower salinity.

In columnar ice below the freeboard, the air volume fraction shows a clear relationship with the brine volume fraction and
temperature (Fig. 6 and Fig. 9b,c,d). Bubbles in columnar ice are often enclosed in brine inclusions (Light et al., 2003), so it is
not surprising that their volumes are linked. Biogeochemical work on the aqueous-gaseous equilibrium in sea ice (Zhou et al.,
2013; Moreau et al., 2014; Crabeck et al., 2019) suggests that the air volume depends on the potential phase change (from
dissolved within brine to gaseous in air bubbles) that takes place in the brine medium. Observations indicate that gas inclusions
shrink or disappear as brine cools (Light et al., 2003; Crabeck et al., 2019). Our observations reveal that during the coldest
months (October to May), the air volume fraction decreased and stayed well below 2 % in FYI and SYI (Fig. 6c). As the brine
inclusions shrink under cooling, the air bubbles contained within them also shrink under the compressive force of the freezing
pressure present in a closed pocket (Crabeck et al., 2019). This freezing pressure results from the gradual volume expansion
linked to the transformation of liquid water to solid ice in closed brine pockets as temperatures fall, which compresses the
trapped bubble and pushes gas into the solution. By the end of winter, brine solutions are heavily concentrated with dissolved
gas (Zhou et al., 2013; Crabeck et al., 2016, 2019).

As the ice warms and the brine volume fraction increases, the air volume fraction also increases (Fig. 6, Fig. 9b,c). Previous
studies have reported that columnar sea ice is typically depleted in air volume fraction, with gas volume less than 2 %, and
that most air voids are found in the ice above the waterline—because below the freeboard, these voids would be filled with
underlying seawater or meltwater (Tsurikov, 1979). However, our observations reveal a notable increase in air volume in
columnar ice even below the freeboard as the ice warmed and became fully permeable (Fig. 7). A similar increase in air
volume fraction during the melting period has also been reported by Wang et al. (2020). Light et al. (2003) observed that
warming brine inclusions lead to bubble enlargement, while Crabeck et al. (2019) reported the appearance of new bubbles in



brine due to nucleation during warming. They related these bubbles' enlargement and nucleation to internal melt within the
brine pockets. In our dataset, the increase in air volume fraction in columnar ice below the freeboard is particularly significant
when the ice temperature is above -2°C (Fig. 9c,d). Above -2°C, the ice undergoes a heavy internal melt, characterised by
the gradual transformation of ice into liquid brine. Tsurikov (1979) and Perovich and Gow (1996) suggested that the density
difference of 10 % between ice and liquid during melting would result in the formation of voids filled with water vapour in
sea ice. For every 1 ml of ice that melts into brine, approximately 0.1 ml of voids is created due to the density difference.
In SYI, the increase in air volume fraction closely follows this relationship (i.e., dotted line in Fig. 9e), and the increase is
especially strong at the onset of pure ice melt close to 0°C (Fig. 9f). For FYI, our observations show that the increase in air
volume is significantly higher than the 10 % increase associated with ice melting (Fig. 9b). Crabeck et al. (2019) explained
that voids formed by internal ice melt create a vacuum in the brine pockets, causing dissolved gases heavily concentrated in
the liquid brine to nucleate into new bubbles and migrate to the gas phase, thus enlarging the bubble volume. This process is
typical of FYI, which is more enriched in dissolved solutes (i.e., salt and dissolved gases) than SYI, which is depleted in salts
and other solutes. Additionally, drained brine pockets can become filled with air even below the freeboard, as shown in Fig. 9
(smallest red circles), the most desalinized layers have the highest air volume fraction. Finally, it is important to note that while
salts are efficiently rejected downward by brine drainage and flushing, gas tends to remain in the ice. Once gas nucleates into
air bubbles, these bubbles are either trapped in the ice or move upward by buoyancy, leading to the accumulation of bubbles
throughout the season.

We demonstrated that the seasonal evolution of sea-ice density is controlled by the temperature-dependent evolution of its
air volume. As ice warms, the air volume within the ice column increases, reducing sea-ice density (Fig. 6d,g). Conversely,
warming also increases brine volume, which slightly counteracts this effect by increasing sea-ice density. However, the impact
of air on density is roughly 10 times greater than that of brine. This, a 4.8 % increase in air volume during winter-to-summer
transitions has the same effect on density as a 43 % decrease in brine volume. During the melt season, changes in sea-ice
density may have a more significant impact on ice freeboard than changes in ice thickness, complicating the interpretation of
ice melt from altimetry data.

## 4.4 Accounting for melt pond drainage

The presence of melt ponds complicates ice thickness retrieval from the freeboard measurements by introducing uncertainties
related to the unknown depth of melt ponds (Landy et al., 2022). Freeboard measurements, including those obtained from ice
coring, are often conducted in areas not covered by melt ponds. Before melt pond drainage, these ponds increase the effective
sea-ice density and lower the freeboard by adding hydraulic head (water height above sea level) (Eicken et al., 2002). After
melt pond drainage, while the average effective sea-ice density and freeboard remain unchanged, unponded areas around melt
ponds experience an uplift (Fuchs et al., 2024). In this section, we investigate the effect of melt ponds and their drainage on the
freeboard of ponded and unponded ice. Our goal is to evaluate the biases in sea-ice density estimates from coring freeboard
measurements.



We observed an increase in FYI freeboard of 0.07 m at the coring site between 6–13 July. This increase occurred right after the drainage of melt ponds on 11–13 July (Webster et al., 2022) and the initial appearance of under-ice meltwater layers (Salganik et al., 2023c). The estimated effective sea-ice density, derived from the coring freeboard, decreased by 8 %, corre-
sponding to a change in freeboard due to a 0.08 m$^3$/m$^2$ loss of meltwater unit volume (Fig. 5a,b). Salganik et al. (2023c), using underwater sonar surveys near the FYI coring site, estimated that under-ice meltwater had an areal coverage of 20 % and a thickness of around 0.5 m. The rapid transfer of this meltwater volume from surface ponds to under-ice layers resulted in a 0.1 m$^3$/m$^2$ decrease in meltwater unit volume. This estimate aligns with the observed ice freeboard uplift from coring, as well as the 0.1 m difference between the sonar draft change and the corresponding ice melt from coring during 6–13 July (Fig. 5a,d).
Transect measurements provide near-daily data on melt pond depth and coverage, although the hydraulic head of melt ponds is usually unknown. Fuchs et al. (2024) used photogrammetry to obtain melt pond depth and freeboard data from three scans on 30 June, 17 and 22 July. These measurements indicated nearly constant melt pond coverage of 22–24 %, with melt pond depth of 0.23±0.28 m, 0.17±0.16 m, and 0.18±0.21 m, and melt pond freeboard of 0.13 m, 0.06 m, and 0.02 m, respectively, suggesting gradual melt pond drainage in July. Transects on 10 July showed a similar melt pond depth of 0.16±0.11 m and
areal fraction of 20 %, just before the drainage event on 11–13 July (Webster et al., 2022).

To assess the impact of melt ponds on sea-ice density estimates, we combined data on melt pond fraction and depth from the transects with the ice thickness and density from the FYI coring site. The presence of undrained melt ponds had a minor effect on ice freeboard. Eq. (6) indicates that undrained melt ponds result in an increase in the effective sea-ice density of 2–6 kg m$^{-3}$. Following melt pond drainage, Eq. (4) and (5) suggest no change in the effective sea-ice density for combined
ponded and unponded areas with substantially different freeboard values. Meanwhile, the estimated increase in freeboard for unponded areas surrounding drained melt ponds was 0.03–0.04 m. This leads to a decrease in effective sea-ice density from 902–907 kg m$^{-3}$ for undrained ponds to around 880 kg m$^{-3}$ for drained melt ponds (Fig. 5b). The estimated 15–23 kg m$^{-3}$ difference in effective sea-ice density for all and only unponded ice areas is significant, as ice coring was conducted in unponded areas. This bias towards coring in unponded areas at some distance from the melt ponds, to avoid artificial drainage, may explain
the lower values of sea-ice density estimated from coring freeboard and draft measurements observed after melt pond drainage. We cannot fully account for the 0.07 m ice uplift observed at FYI coring site, which could be attributed to substantial meter-scale spatial variability in sea-ice freeboard (Fig. 8a,d). On a larger scale, pond drainage likely caused a 0.03 m ice uplift for CO2 (Fig. 3b) and a 0.05 m uplift for the full ALS scan (Fig. 3c), aligning with our analytical estimates. The varying times of pond drainage across different ponds (Webster et al., 2022) complicate the analysis of hydrostatic balance at the floe scale.
Eq. (4) indicates that the effective sea-ice density for areas with drained melt ponds is higher than for pre-melt sea ice without melt ponds. This effect, due to the higher effective density for uneven surfaces, may partially explain why freeboard and draft measurements yield slightly higher sea-ice density than hydrostatic weighing throughout the year (Fig. 5b).

In this section, we evaluated the potential effect of both drained and undrained melt ponds on estimates of the effective sea-ice density. Considering the presence of melt ponds helps reduce biases in freeboard measurements conducted partially
(like ALS) or fully (like coring) at sea ice without melt ponds. We also demonstrated that melt pond drainage causes an uplift of unponded ice, observable on different scales. The magnitude of this uplift can be described analytically, although small-





scale freeboard observations from coring are subject to significant spatial variability, making measurements of such uplift less accurate.

### 4.5 Impact of ridges and snow thickness spatial variability

The presence of ice ridges can significantly affect snow freeboard values, primarily due to the thicker snow that accumulates above deformed ice (Itkin et al., 2023). For instance, in May, the ALS snow freeboard was 0.16 m larger at a heavily deformed CO2 area compared to a smoother surrounding area (Fig. 5b,c). To improve the intercomparison of freeboard evolution using different methods, we categorized all ALS freeboard measurements into level ice and ridge classes, using a threshold derived from the co-located observations of a ridge freeboard and draft from ALS and sonar. In the summer, we found that the CO2 had

an areal fraction of ridge keels of 32±5 %, which was larger than the 22±4 % ridge keel fraction for the full ALS scans (with 3 times smaller areal fraction of ridge sails). Similarly, Itkin et al. (2023) estimated a ridge keel fraction ranging from 22 % to 45 % for CO1 using electromagnetic sounding during winter and spring. The summer evolution of ridge freeboard is more complex than of level ice (Salganik et al., 2023e), partially due to delayed snow melt. Although ridges were undersampled compared to level ice (with only four sampling events), the temporal evolution of ridge density was similar to FYI (Fig. 6g),

as well as the freeboard evolution of both ridge and level ice during melt season (Fig. 3b,c). Ridges may have a smaller air fraction increase than FYI (Fig. 6f) as they were substantially colder than level ice in summer (Lange et al., 2023). The main importance of ridges for ice thickness retrieval from snow freeboard measurements lies in their ability to increase surface roughness, which leads to the accumulation of substantially thicker snow above them (Itkin et al., 2023). This complicates the upscaling of ice thickness retrieval from snow freeboard, as snow thickness can vary significantly in areas with different

ridge fractions. Therefore, the pre-melt estimates of ice thickness from ALS snow freeboard measurements outside of CO1 are uncertain, as snow thickness was only measured within CO1 and CO2, which have unrepresentative ridge fractions.

Snow thickness is a major source of uncertainty for estimates of sea-ice density derived from airborne snow freeboard measurements outside of the melt period (Landy et al., 2020; Jutila et al., 2022). The spatial variability of snow thickness can be as large as the variability of sea-ice freeboard with different densities. While our summer observations are characterized by

a thin and homogeneous SSL (Webster et al., 2022), snow thickness in winter and spring shows large spatial variability due to the differences in surface roughness among various types of ice. Itkin et al. (2023), using transect measurements, demonstrated that the CO2 area, with approximately equal areal fractions of level ice, deformed ice, and ridges, had average snow thicknesses of 0.16 m, 0.26 m, and 0.40 m, respectively, in early May, with standard deviations ranging from 0.12 m to 0.20 m. In early May, snow depth measurements from IMBs showed 0.16±0.02 m for level FYI (3 buoys) and 0.23±0.07 m for level SYI (10

buoys). Snow depth at coring sites in January–May averaged 0.13±0.04 m for 8 FYI coring events and 0.14±0.03 m for 4 SYI coring events (Fig. 6a). These measurements illustrate the substantial spatial variability of snow thickness above level ice within CO1, obtained using various methods. Eq. (3) indicates that the uncertainty in sea-ice density is 5–10 kg m$^{-3}$ per 0.01 m of snow for a known snow freeboard and 1–3 kg m$^{-3}$ per 0.01 m of snow for a known ice freeboard. The observed seasonal variability of FYI density, which ranges around 30–60 kg m$^{-3}$ (Fig. 5b), corresponds to snow thickness estimate uncertainties



of 0.04–0.08 m. These uncertainties are comparable to standard deviations of snow thickness measured by IMBs, coring, and transect.

Snow thickness can also vary on a kilometer scale. ALS measurements in April–May showed 0.12–0.17 m larger snow freeboard values and 1.5 times larger ridge areal fraction at CO2 compared to the surrounding area (Fig. 3b,c), despite comparable ice thickness from IMBs and electromagnetic sounding (von Albedyll et al., 2022). Our estimate of the areal ridge fraction—32 % for CO2 and 22 % for the full ALS scans—assuming 2 m level ice thickness from IMBs and a typical ridge thickness of 6 m (Strub-Klein and Sudom, 2012) would explain only a 0.03 m difference in the average snow freeboard between CO2 and the ALS full scans, with the remaining freeboard difference attributed to variations in snow thickness. For level ice, the ALS snow freeboard was 0.10–0.14 m larger for CO2, which is comparable to the absolute values of the observed level ice freeboard. This indicates significant uncertainties in sea-ice density estimates using pre-melt snow freeboard measurements due to substantial spatial variability of snow thickness on different scales, unless such measurements are co-located with snow measurements, as in Jutila et al. (2022). Moreover, snow thickness uncertainties cannot be simply mitigated by using larger spatial coverage. On kilometer scales, snow thickness also strongly depends on sea-ice roughness and ridge fraction.

## 5 Conclusions

In this study, we use weekly observations of first- and second-year ice density from hydrostatic weighing. During the melt season, the bulk density of first-year ice decreased from 910 kg m$^{-3}$ to 880 kg m$^{-3}$, while the bulk density of second-year ice showed a smaller decline from 912 kg m$^{-3}$ to 905 kg m$^{-3}$. The observed seasonal changes in sea-ice density were primarily due to the rapid increase of air volume fraction during the melt season. For first-year ice, the bulk air volume fraction rose from 1 % to 6 %, and for second-year ice, it increased from 1 % to 2 %. Our seasonal dataset indicates that the previous assumption that columnar sea ice below the freeboard has a depleted air volume fraction (i.e., less than 2 %) is no longer valid. The substantial increase in air volume fraction across the whole ice column during the melt season significantly affects sea-ice buoyancy and its freeboard. The increase in air volume is strongly related to two factors: (1) internal melt, which creates voids, enlarges bubbles, and nucleates new bubbles, and (2) the replacement of liquid brine by air in drained inclusions.

The measurements of first-year ice density were validated using weekly ice freeboard and draft measurements from ice coring, as well as co-located snow freeboard measurements from an airborne laser scanner and ice draft measurements from underwater sonar. Both methods showed a comparable decreasing trend in first-year ice density, similar to the direct density measurements from hydrostatic weighing. The estimates of pre-melt first-year ice density in March–May from co-located laser scanner and sonar measurements were higher, at 926±4 kg m$^{-3}$, compared to 912±2 kg m$^{-3}$ from weighing. This discrepancy may be attributed to the uncertainties in snow thickness for estimates using scanner measurements, the effect from ice surface roughness, or potential brine losses during weighing.

The decrease in sea-ice density during melt season leads to non-decreasing values of its freeboard, which complicates estimates of sea-ice melt from altimetry measurements. During June–July, we measured the total melt of level ice to be 0.6–0.7 m using ice coring, underwater sonar, and ice mass balance buoys. Despite the absence of snow above level ice in July, this ice



loss was accompanied by an increase in first-year ice freeboard of 0.02 m at the coring site and at the area of the co-located laser scanner and underwater sonar observations. For the whole 0.9 km$^2$ investigated ice floe, the level ice freeboard also increased by 0.03 m, while the freeboard of the surrounding 50–60 km$^2$ experienced an increase of 0.01 m in level ice areas. Our study underscores the necessity to account for seasonal changes in sea-ice density, particularly the air volume fraction, for more accurate ice thickness retrievals.

*Data availability.*   All datasets used in this study are publicly available. The FYI, SYI, and ridge salinity, temperature, and density are available in Oggier et al. (2023a, b); Salganik et al. (2024, 2023d), the airborne laser scanner measurements can be found in Hutter et al. (2023a), the multibeam sonar data can be found in Katlein et al. (2022), the Magnaprobe snow and melt pond depth measurements can be found in Itkin et al. (2021), the snow density data can be found in Macfarlane et al. (2021), the ice mass balance buoy data can be found in Lei et al. (2021) and Salganik et al. (2023a), the helicopter-borne RGB orthomosaics can be found in Neckel et al. (2023), melt pond bathymetry is available in Fuchs and Birnbaum (2024), and the core hydrographic data can be found in Schulz et al. (2023).

*Author contributions.*   ES and OC contributed to the design of the study. ES, NH, and NF collected and processed the field data. ES and OC undertook the statistical analyses and interpreted the results. ES and OC prepared the manuscript with contributions from all co-authors.

*Competing interests.*   The authors declare that no competing interests are present.

*Acknowledgements.*   The work carried out and the data used in this paper are part of the international Multidisciplinary drifting Observatory for the Study of the Arctic Climate (MOSAiC) with the tag MOSAiC20192020. We thank all persons involved in the expedition of the Research Vessel *Polarstern* (Alfred-Wegener-Institut Helmholtz-Zentrum für Polar- und Meeresforschung, 2017) during MOSAiC in 2019–2020 (Project_ID: AWI_PS122_00) as listed in Nixdorf et al. (2021). We especially acknowledge Mats A. Granskog, Marcel Nicolaus and Donald Perovich for their efforts in coordinating the sea-ice physics work during MOSAiC. We are also grateful to Marcel Nicolaus for his efforts in coordinating the ROV work during MOSAiC, and to Christian Katlein for processing ROV multibeam sonar data.

Evgenii Salganik and Jack Landy were supported by Research Council of Norway project INTERAAC (grant no. 328957). Odile Crabeck was supported by the FRS-FNRS (Research Credit MOSAiC J.0051.20 and Research Project Sea Ice Spray - T.0061.23). Odile Crabeck was also supported by the FRS-FNRS Fellowship (grant 1.B.103.21F) and GreenFeedBack (Greenhouse gas fluxes and earth system feedbacks) funded by the European Union's HORIZON research and innovation program under grant agreement No. 101056921. ROV operations were jointly supported by UKRI Natural Environment Research Council (NERC) and the German Federal Ministry of Education and Research (BMBF) through the Diatom ARCTIC project (BMBF grant no. 03F0810A). Data processing as well as the position of Nils Hutter was funded by German Federal Ministry of Education and Research (BMBF) project IceSense — Remote Sensing of the Seasonal Evolution of Climate-relevant Sea Ice Properties (03F0866A). Nils Hutter was partially funded by the Cooperative Institute for Climate, Ocean, & Ecosystem Studies (CICOES) under NOAA Cooperative Agreement NA20OAR4320271, Contribution No. 2023-1311. Niels Fuchs acknowledges



funding from the BMBF project NiceLABpro (03F0867A) and from the Deutsche Forschungsgemeinschaft under Germany's Excellence Strategy (EXC 2037; CLICCS – Climate, Climatic Change, and Society; project no. 390683824). Philipp Anhaus was supported by BMBF through the Diatom ARCTIC project (BMBF grant no. 03F0810A) and the IceScan project (BMBF grant no. 03F0916A). Views and opin-620  ions expressed are however those of the author(s) only and do not necessarily reflect those of the European Union or CINEA. Neither the European Union nor the granting authority can be held responsible for them.



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
