# Peer review of "Impacts of air fraction increase on Arctic sea-ice thickness retrieval during melt season"

_EGUsphere, 2024_

## Referee Comment (RC1)

**Peer-review:** "Impacts of air fraction increase on Arctic sea-ice thickness retrieval during melt season" by Evgenii Salganik, Odile Crabeck, Niels Fuchs, Nils Hutter, Philipp Anhaus, and Jack Christopher Landy

**General comments:**

This paper analyses the importance of accurately estimating the air volume in density measurements of sea ice. The importance of the air fraction on sea ice density is not recognized in the current literature. The paper compares different methods to estimate the sea ice density, i.e., hydrostatic weighing and indirect calculation from ice freeboard and ice thickness measurements. The paper uses data from different publicly available data sets, which cover both first-year ice and second-year ice. The paper discusses the influence of melt ponds, ridges, and snow thickness variability on density measurements and freeboard measurements. This research is relevant because sea-ice density estimates affect ice thickness based on freeboard measurements using remote sensing.

The findings of the paper are relevant and appropriate for the journal.

My recommendation for this paper is "major revision." This is not a reflection of the quality of the research, but only of the presentation of the findings. The paper is long and difficult to read. It is not immediately clear what the main aim of the paper is. The paper introduces topics that are adjacent to the research, and it is not always clear how they relate to the main research aim. The actual research becomes lost as a result. Considering that this paper aims to target the remote sensing community (to my understanding), the paper would benefit from narrowing down its scope and from focusing its arguments to make the paper more accessible.

That being said, the overall language is clear. The figures are clear and add to the understanding of the paper. The paper could benefit from tables that summarize data that is currently only presented in the text. Furthermore, the authors should pay attention that each paragraph of text addresses only one topic at a time. Many paragraphs in the results and discussion sections are exceedingly long and jump between multiple different topics, which makes the paper very difficult to read.

**Major comments:**

1. The introduction is quite long, which dilutes the main message of the paper. The need of the research is introduced in the first two paragraphs of the paper. The topics "sea ice density" and "air in ice" are only introduced in the fifth paragraph (Lines 71-97). It is not directly clear how the remaining paragraphs support the aim of the paper. The information in these paragraphs could possibly be cited in the Discussion section when the information is needed or unexpected to the relevant research community.

2. The results section is difficult to read. Readability could be improved by summarizing relevant data in tables that the authors could refer to. Section 3.4 is especially difficult to read because each paragraph presents multiple ideas at the same time. Readability could be improved by rewriting this section to include smaller paragraphs that each focus on one idea at a time.

3. The discussion is the most difficult to read of all the sections and requires extensive restructuring and refocusing. The current discussion section is exceedingly long (8 out of 23 pages). Moreover, the discussion section presents an extensive overview of literature as well as additional analysis of the results. The discussion introduces topics that do not necessarily support the aim of the research paper. Many of the paragraphs are very long and present multiple different ideas at the same time. Please consider restructuring the text into shorter paragraphs that are focused on one topic each. Please also refer to a Section/Figure/Table instead of repeating the data in the discussion.

4. The conclusion clearly states the aim of the paper and summarizes individual results nicely. However, a conclusion should not merely summarize the results; it should also interpret the findings of the paper at a higher level of abstraction.

**Minor comments:**

1. The title is appropriate, but it could be improved. The paper mainly focuses on the importance of accurate density measurements, and air fraction increase is presented as one variable affecting density. Density estimates affect sea-ice thickness retrieval; however, this message becomes lost in the paper. Sea-ice thickness retrieval is presented as the reason why this paper is important, but not as the main aim of the paper. The title may not need to be changed depending on how the authors decide to re-structure the remaining paper.

2. The abstract captures the essence of the hypothesis, findings, and significance. Lines 1-7 are clear and support the title of the abstract. Lines 8-11 present a discussion of the results and Lines 11-13 highlight the relevance, without mentioning the air fraction nor density. The coherence between these different sections is not clear and the scope of the abstract could be narrowed down.

3. Lines 167-172: This paragraph explains the role of meltwater and how to account for meltwater using methods that are not related to ice coring or mass balance buoys. You then refer to Section 2.3. Could you highlight how this paragraph connects with the remaining Section 2.1?

---

## Author Comment (AC4)

[Figure]

Figure RC4-1: Modal (left) and mean (right) freeboard for full ALS surveys during the freezing season.

[Figure]

Figure RC4-2: Snow/SSL freeboard change between 10 May and 30 June (a), 17 July (b), and 22 July (c).

---

## Author Response (AR1)

***Reviewer RC1, Alice Petry***

We thank the reviewer, Alice Petry, for their helpful comments and the work they put into the review. Our answers start with "Reply" in bold, *following the original reviewers' comments in italics*.

*General comments:*

*This paper analyses the importance of accurately estimating the air volume in density measurements of sea ice. The importance of the air fraction on sea ice density is not recognized in the current literature. The paper compares different methods to estimate the sea ice density, i.e., hydrostatic weighing and indirect calculation from ice freeboard and ice thickness measurements. The paper uses data from different publicly available data sets, which cover both first-year ice and second-year ice. The paper discusses the influence of melt ponds, ridges, and snow thickness variability on density measurements and freeboard measurements. This research is relevant because sea-ice density estimates affect ice thickness based on freeboard measurements using remote sensing.*

*The findings of the paper are relevant and appropriate for the journal.*

**Reply:** Dear Alice, we thank you for your positive evaluation of our manuscript. We have done our best to address your main concerns, including a substantial shortening of the manuscript, adding tables with results, and refining paragraph structure. We have provided detailed replies to all your questions and suggestions below, attempting to provide enough referenced evidence.

*My recommendation for this paper is "major revision." This is not a reflection of the quality of the research, but only of the presentation of the findings. The paper is long and difficult to read. It is not immediately clear what the main aim of the paper is. The paper introduces topics that are adjacent to the research, and it is not always clear how they relate to the main research aim. The actual research becomes lost as a result. Considering that this paper aims to target the remote sensing community (to my understanding), the paper would benefit from narrowing down its scope and from focusing its arguments to make the paper more accessible.*

**Reply:** We agree with your suggestion and significantly shorten our manuscript from 600 to less than 470 lines, with the main shortening related to introduction and discussion sections.

The aim of this study, described as early as in its abstract, is to present observations of increasing air fraction in sea ice during summer melt and its effects on sea-ice density and freeboard. It includes a detailed validation and upscaling of density estimates using direct and indirect methods covering different spatial scales. Finally, it attempts to explain the observed air fraction and density evolution and connect it to the thermodynamic state of sea ice (salinity and temperature). To make it clearer, we modified the title of our study.

Some of the themes covered in discussion and probably referenced as "adjacent to the research" may not immediately look relevant to the estimates of sea-ice density. Meanwhile, we provided a detailed explanation and references why melt ponds and ridges are important for density estimates, especially in remote sensing context. In brief, melt ponds are substantially complicating the hydrostatic balance of summer sea ice, and without accounting for their evolution, the evolution of ice density obtained from ice freeboard cannot be linked purely to the change of ice air fraction. It was vital to show that the observed ice uplift was linked to density evolution, not to melt pond drainage. Similarly, the known effect of ridges on snow thickness complicates estimates of ice freeboard from the measured snow freeboard, which may substantially decrease the accuracy of density estimates from ice hydrostatic balance. As we focus on density of underformed ice, introduction of ice classification is needed. In addition, as we cover the pre-melt season with thick snow cover as a reference to summer

observations, we must highlight limitations related to snow thickness variability on different scales. Meanwhile, we removed most unnecessary details from the discussion sections about ridges and melt ponds, from 85 to 50 lines.

Finally, we indeed believe that our paper primarily focused on the remote sensing community, aiming to connect sea-ice physical observations on different scales, remote sensing methods, and the geochemistry of sea ice. We attempted to show that small-scale changes in sea-ice air and brine composition are (1) driven by physical parameters such as temperature and salinity (2) and affect the accuracy of ice thickness retrievals from its freeboard. We added a more detailed motivation for why certain parts of the discussion are present.

We agree with most of your concerns and made substantial changes related to the study's length, focus, and readability.

*That being said, the overall language is clear. The figures are clear and add to the understanding of the paper. The paper could benefit from tables that summarize data that is currently only presented in the text. Furthermore, the authors should pay attention that each paragraph of text addresses only one topic at a time. Many paragraphs in the results and discussion sections are exceedingly long and jump between multiple different topics, which makes the paper very difficult to read.*

**Reply:** We agree with your suggestion and substantially changed the structure of longer paragraphs to limit their scope. We also added a table summarizing the results of first-year ice sampling, while also removing a few exact values in the text when necessary. Finally, we reduced the study length by 20% from 600 to less than 470 lines.

Major comments:

*The introduction is quite long, which dilutes the main message of the paper. The need of the research is introduced in the first two paragraphs of the paper. The topics "sea ice density" and "air in ice" are only introduced in the fifth paragraph (Lines 71-97). It is not directly clear how the remaining paragraphs support the aim of the paper. The information in these paragraphs could possibly be cited in the Discussion section when the information is needed or unexpected to the relevant research community.*

**Reply:** We removed part of the introduction, describing snow and melt pond evolution (lines 98–106). Meanwhile, we cannot agree with the suggestion that only lines 15–49 and 71–97 should be kept in this section. Introduction should not only include motivation of the study (lines 15–49, the first two paragraphs), but also the current understanding of the addressed question.

Second, the need for sea-ice density for thickness retrieval was mentioned as early as in line 22. Its direct implications are introduced in line 33. An overview of existing density datasets is presented in lines 57–70. Indeed, air fraction is introduced only in line 71. We think that without a proper introduction of (1) why density is important and (2) limited measurements of ice density, especially during melt, we cannot start a discussion about the air fraction of sea ice.

We cannot fully agree that the information given in lines 50–70 does not support the aim of this study. The importance of air fraction should be justified before this niche topic is discussed. In lines 50–56, we briefly present the main methods of sea-ice density measurements, their accuracy, and their limitations. In the following lines 57–70, we present an existing knowledge of density ranges and seasonality, with a focus on potential strong density variations during melt season, as well as confirming substantial observational gaps in density measurements during summer and autumn. Finally, lines 107–115 give a short summary of the whole study. We do not think that presenting

measuring techniques and historical observations of sea-ice density dilutes the main message of the paper. Meanwhile, we reduced the introduction by half from 100 to less than 60 lines.

*The results section is difficult to read. Readability could be improved by summarizing relevant data in tables that the authors could refer to. Section 3.4 is especially difficult to read because each paragraph presents multiple ideas at the same time. Readability could be improved by rewriting this section to include smaller paragraphs that each focus on one idea at a time.*

**Reply:** We followed your suggestion and summarized our estimates of the main ice physical properties in Table 1, presenting bulk FYI density (from three methods), as well as salinity, temperature, freeboard, snow and ice thickness, air and brine volumes. For most of our results, we prefer to show our results in figures, as tables are convenient if sampling from different methods was performed in parallel (which is not the case for our methods). Therefore, we only added a single table for sampling at the coring and ROV sites. We divided Section 3.4 into two subsections covering (1) seasonal evolution of air volume and density and (2) similarities between first- and second-year ice. We also reduced its length from 41 to 25 lines.

*The discussion is the most difficult to read of all the sections and requires extensive restructuring and refocusing. The current discussion section is exceedingly long (8 out of 23 pages). Moreover, the discussion section presents an extensive overview of literature as well as additional analysis of the results. The discussion introduces topics that do not necessarily support the aim of the research paper. Many of the paragraphs are very long and present multiple different ideas at the same time. Please consider restructuring the text into shorter paragraphs that are focused on one topic each. Please also refer to a Section/Figure/Table instead of repeating the data in the discussion.*

**Reply:** We reduced the size of the discussion from 220 to 180 lines. We also introduced a substantial restructuring of separate paragraphs.

Our updated discussion restates the results (section 4.1 "Importance of..."), compares them with satellite observations (section 4.2 "Comparison with radar altimetry..."), compares them with previous observations (section 4.3 "Comparison with..."), compares methods used in our study and in other studies about sea-ice density (section 4.4 "Comparison of methods..."), provides an explanation and interpretation of the observed results (section 4.5 "Geochemistry..."), and discusses limitations of indirect methods, which in our case were related to melt ponds (section 4.6 "Accounting for melt ponds") and ridges and snow thickness variability (section 4.6 "Impact of ridges..."). We believe that all these six sections are relevant and novel for the topic of summer sea-ice density. Meanwhile, we tried to remove all unnecessary and repeating statements in discussion.

Second, a discussion typically should include a comparison of presented results and previous observations (of ice density, in our case). Meanwhile, we significantly reduced the length of Section 4.3.

Third, in the section about sea-ice geochemistry, we present a detailed interpretation of what has been observed and which processes can explain these results. It indeed references previous findings in ice geochemistry, which was vital to provide a physics-based explanation of the observed seasonal changes in air volume.

Fourth, we have already listed the reasons for which each of the six sections was included in the discussion.

We agree with the comment about the length of individual paragraphs and attempted to make them shorter and more focused. We also added a few references to previously presented sections and figures.

*The conclusion clearly states the aim of the paper and summarizes individual results nicely. However, a conclusion should not merely summarize the results; it should also interpret the findings of the paper at a higher level of abstraction.*

**Reply:** We agree with your suggestion and added a concluding paragraph, suggesting that future ice density parametrization should be more focused on the air volume evolution governed by ice temperature: "We showed that both our and historical observations of sea-ice density reveal its strong dependence on sea-ice temperature and salinity, with the range of summer density decrease potentially making around half of typical summer thickness loss not detectable from ice freeboard observations."

*Minor comments:*

*The title is appropriate, but it could be improved. The paper mainly focuses on the importance of accurate density measurements, and air fraction increase is presented as one variable affecting density. Density estimates affect sea-ice thickness retrieval; however, this message becomes lost in the paper. Sea-ice thickness retrieval is presented as the reason why this paper is important, but not as the main aim of the paper. The title may not need to be changed depending on how the authors decide to re-structure the remaining paper.*

**Reply:** We changed our title to "Impacts of air fraction increase on Arctic sea-ice density, freeboard, and thickness estimation during melt season" to broaden the description of the performed analysis. We think that any study providing novel findings should provide a thorough validation of the presented results, especially in geophysics, where spatial variability often plays an important role. In our case, the main reason for validation, intercomparison, and upscaling is to support our measurements of air volume increase, which mainly governs seasonal evolution of sea-ice density.

We cannot fully agree with the statement that the topic of ice thickness retrieval is lost in the paper. A substantial part of the paper presents observations of ice freeboards, a parameter mainly used by remote sensing methods for ice thickness retrieval. Freeboard observations are often presented together with ice draft measurements, directly connecting ice freeboard and thickness, which is equivalent to the validation of ice thickness retrieval. Nearly every figure in our study (Fig.1,3-6,8) presents values of sea-ice freeboard and ice thickness or thickness change. Meanwhile, we substantially reduced the length of discussion.

Finally, one of the key messages of this study is that ice thickness decrease cannot be captured from ice freeboard observations without considering seasonal evolution of ice density. We think that this connection, which we focused on in most parts of our study, is the essence of ice thickness retrieval.

*The abstract captures the essence of the hypothesis, findings, and significance. Lines 1-7 are clear and support the title of the abstract. Lines 8-11 present a discussion of the results and Lines 11-13 highlight the relevance, without mentioning the air fraction nor density. The coherence between these different sections is not clear and the scope of the abstract could be narrowed down.*

**Reply:** We cannot agree that lines 11–13 do not mention density, as it concludes as «underlines the importance of considering density changes in retrieval algorithms». Meanwhile, we modified the concluding sentence of our abstract, mentioning the importance of air fraction on ice thickness retrieval during melt season.

*Lines 167-172: This paragraph explains the role of meltwater and how to account for meltwater using methods that are not related to ice coring or mass balance buoys. You then refer to Section 2.3. Could you highlight how this paragraph connects with the remaining Section 2.1?*

**Reply:** We agree with your suggestion and moved this paragraph directly to Section 2.3 to avoid repetition.

*Reviewer RC2:*

We thank the reviewer for their helpful comments and the work they put into the review. Our answers are starting with "Reply" in bold, following *the original reviewers' comments in italics*.

*The manuscript uses sea-ice thickness measurements from different methods to understand the influence of sea-ice density on these measurements. Especially, the influence of air fraction within the ice is highlighted. Ice core measurements are compared to other methods estimating the sea-ice density. The main conclusion is that the air fraction is one of the main contributions to the density change in the ice regarding the change in seasons. A better understanding of sea-ice density and its seasonality potentially improves sea-ice thickness retrievals from satellite altimetry.*

*Overall, I think the manuscript makes a clear contribution to the understanding of how sea-ice thickness measurements are influenced by the smaller scale properties of the sea ice.*

**Reply:** Dear reviewer, we thank you for your positive and constructive evaluation of our manuscript. We have done our best to address your main concerns. We have provided detailed replies to all your questions and suggestions below, attempting to provide enough referenced evidence.

*Major Comments:*

*While the scientific advances within the manuscript are present, I found it sometimes complicated to follow the specific contributions of this manuscript. I suggest revision in the following areas to improve the clarity of the manuscript:*

*The manuscript often deviates from its main topic. Judging the manuscript on its title, I assume the focus should be on air-fraction in ice and density changes and its impact on sea-ice thickness retrievals, but not on the comparison of retrieval products itself. Additionally, I assume that the freeboard products in and their comparison with each other do not carry novelty for this manuscript as I did not find any mention of their novelty and all data products seem to be published elsewhere.*

**Reply:** We substantially reduced the length of our study by removing topics that are less relevant for the main message of our study.

Indeed, a substantial part of it is devoted to the validation of small-scale direct measurements of sea-ice density using other more often used indirect methods based on ice hydrostatic balance and freeboard measurements. The estimate of a parameter that does not have a generally accepted parametrization or model to predict it, such as air volume or sea-ice density, should be validated. The seasonal changes of sea ice density presented in our study can be only explained by an increase in air volume fraction, which was directly observed using weighing and indirectly using various freeboard and draft measurements. As all the existing methods of sea ice density and air volume estimates have known flaws, it was vital to make a comparison of both density estimates and corresponding freeboard evolution. Snow and ice freeboard is the prime method of sea-ice thickness retrieval, as such data is much easier to obtain. Meanwhile, validation of our density estimates was mainly performed to support our measurements of air volume seasonal evolution.

We are unsure what is meant by the lack of novelty in freeboard data products. Indeed, all the used datasets, mainly collected, processed, and published by a few co-authors of this study, are publicly available as required by both MOSAiC and EGU policies. MOSAiC was an interdisciplinary research program, where the data is aimed to be used by the whole community and was collected with the help of many people. Therefore, we do not feel any need to the underline novelty of ice density, freeboard, melt pond depth, and ice bottom topography datasets despite being significantly

responsible for their publication. We are also not aware of any studies that used this specific data (coring and airborne freeboard, multibeam sonar draft, and photogrammetric melt pond depth) for density estimates during melt season.

*The manuscript seems off balance comparing the length and level of detail in the introduction, results and discussion with each other. Additionally, the whole results section features a lot of detailed numbers, which is very nice to create access to precise numbers, but it makes the text difficult to read. Some information could be combined, e.g., with sentences to highlight that the same behavior across measurement methods is similar (e.g. decreasing in density L273-289 with the values also being represented in Figure 5).*

*The current conclusions section is mainly a summary of the manuscript but does not include remarks on the consequences and impacts of this research. Additionally, it is rather long and, thus, also makes it hard on the reader to identify the specific contribution.*

*Thus, I would suggest a thorough revision to focus on which information is needed for the specific topic of a paragraph and the storyline and which level of detail is needed. The minor comments will also include some sections I found not beneficial to the main theme of the manuscript and thus, would remove or shorten.*

**Reply:** We agree with your suggestion and substantially reduced the length of discussion. We tried to remove unnecessary or repeated materials and to remove some of the presented exact values.

We partially agree with the suggestion to merge some of the estimates of sea-ice density, and we removed the exact values of the effective sea-ice density for ROV site considering melt ponds, only stating that they were slightly lower than estimates without melt pond effect. Some other values, together with estimates of air volume fraction, are the key values, and we prefer to keep the maximum transparency regarding deviations from using different methods. We also added most of the key values from the results section to Table 1.

We added a concluding comment to our conclusions, suggesting the consequences of our study. The Cryosphere guidelines do not include any size limitations, including the length of the conclusions.

Finally, we made corresponding changes related to most of your specific suggestions, with very rare exceptions. And we thank you for providing a detailed description of what can be modified.

*Minor Comments:*

*L22-27: The suggestion is to remove these sentences, as it discusses sea-ice thickness retrievals while the introduction has not arrived at its core theme of sea-ice density and air fraction yet. Additionally, the transition from new last sentence to next paragraph would improve.*

**Reply:** We agree with your suggestion and removed these two sentences about measurements from upward-looking sonars, buoys, and electromagnetic sounding.

*L57-69: It could be beneficial to present this information in a table and then reference to highlights of the values as the presented density values are very close in range and hard to comprehend and compare in text.*

**Reply:** We agree with your suggestion and added all known to us observations of sea-ice density in a separate Figure 1, where first-year ice density is plotted against time and ice temperatures. We believe that this is more illustrative to show both seasonality and lack of data for some seasons and

temperature ranges than compiling them in a table. The figure also helps for future discussion of comparison of our observations with previous ones.

*L71-97: This paragraph is too long and covers too many ideas, e.g., at L78 the topic changes from general introduction to air fraction on sea-ice density to seasonal evolution of density and then changes again around L89 to how gas exits within sea ice and listing of different processes. Additionally, sentence "Given that small …" (L88-89) summarizes the motivation very well, but is buried in a low-stress position.*

**Reply:** We agree with your suggestions and added a new paragraph at line 89. We also separated parts of this paragraph related to (1) the effect of air fraction on density and (2) physical processes during ice seasonal evolution. In addition, we substantially shorten this part.

*L101-104: To shorten the manuscript, the effect of melt ponds on the freeboard change could be removed.*

Reply: We agree with this suggestion and removed the lines 98–106.

*L147-148: Does F1 and F2 stem from Cox and Weeks (1983) and Leppäranta and Manninen (1998) respectively or are two different values for F1 and F2 used each?*

**Reply:** Thank you for the suggestion. We added a clarification that both studies presented the parametrization of F1 and F2 coefficients, but for different temperature ranges. For ice colder than -2 °C in Cox and Weeks (1983) and for ice warmer than -2 °C in Leppäranta and Manninen (1998).

*L153-155: These two sentences feature results, and it is not clear, why they are presented in the method section.*

**Reply:** To address your concern, we removed the corresponding Fig. 2 to decrease the number of presented results in this section. In this paragraph, we attempted to explain why a specific formulation was chosen out of several presented in Cox and Weeks (1983). The authors present several formulations, as it was not well understood whether gas and brine are connected or disconnected. Here we show how different formulations would affect our results and that the difference is much less than seasonal changes. We think it is important to mention such limitations and to prove that they are not affecting the main message of the study, presented in the results. In the current form, we only present the relative difference between two formulations without presenting our results.

*L155: What section is referenced by "the following section"? The results?*

**Reply:** Yes. We clarified that in results we will focus on the selected formulation for air volume estimates to avoid focusing on minor differences between these formulations.

*L163: The citation of Macfarlane et al. (2023) is unclear to me in the way it is implemented in the text. What is from this publication?*

**Reply:** This reference presents both a description of the surface scattering layer (SSL) and measured values of snow and SSL densities. We removed em-dashes and put the SSL description in brackets to make it clearer.

*L165: I suggest using "calculated/estimated" instead of "found".*

**Reply:** We agree with your suggestion and made the suggested changes.

*L167-172: The reference to Section 2.3 could be shortened into one sentence to prevent repetition between the sections.*

**Reply:** We agree with your suggestion and moved this entire paragraph directly to the section of methods "2.3 Melt ponds" to avoid a repetition.

*L216-217: Do you mean that the melt pond is either the only melt pond in the area (unponded) or surrounded by several melt ponds (ponded)? It is hard to understand what exactly drained unponded (L219) is referring to, e.g., how can something drain which does not exist?*

**Reply:** To avoid such questions, we explained that drained unponded ice is ice surrounding drained melt ponds following lines 214–215. This means that after the melt pond drainage, the freeboard of the melt pond becomes different from the freeboard of ice, which surrounds the given melt pond.

*Section 2.3: Is it possible to create a graph/decision tree or logical table for visualizing which assumption belongs to which case?*

Reply: We agree with your suggestion and added the following illustration as Fig.3:

[Figure]

*L238-239: To increase focus within the results sections, the first sentence should distinguish between main results (sea-ice density and air and brine volume) and their impact on thickness retrievals. Guiding data (e.g., ice thickness and freeboard measurements) should be mentioned afterwards.*

**Reply:** We agree with your arguments, and we moved the section presenting ice freeboard and thickness evolution after the sections presenting results of air volume and ice density during melt season.

*L280: The topic of the paragraph switches from comparing the decline in density to the calculation of the pre-melt density. The start of a new paragraph could be considered here.*

**Reply:** We agree with your suggestion and moved the text related to pre-melt estimates to a separate paragraph.

*L288-295: This paragraph could benefit from a topic sentence, a summary sentence and some restructuring: It seems that all (except during the melt pond event?) air and brine fractions increased, but then the comparison time periods vary between comparing Dec – Jun to Jul or May to Jul? Additionally, Figure 5 only shows a period from May to Jul.*

**Reply:** We agree with your suggestion and added a starting sentence summarizing increase of air and brine volumes of FYI and SYI during melt season.

*L302-304: Which parameters are considered to be the sea-ice physical parameters? A topic sentence at the start could explain the reader what to expect.*

**Reply:** We agree with your suggestion and added an introduction sentence that describes which parameters we present here due to their effect on either the hydrostatic balance of sea ice (snow and ice thickness) or sea-ice density (air and brine volume, salinity, temperature).

*L303-309: This paragraph feels very confusing as it jumps between physical parameters (sea-ice temperature, thickness/freeboard, snow thickness, salinity) without guiding the reader to connect them and seasons (e.g., freezing season (Oct – Mar) to spring leading to June (summer?) and then going back to winter).*

**Reply:** This paragraph is needed to give an overview of all parameters influencing sea-ice hydrostatic balance (snow thickness, ice thickness, and freeboard), as well as parameters defining ice bulk density (salinity, temperature, air, and brine volumes). This is presented to give an overview and to avoid questions about additional parameters that can influence ice hydrostatic balance apart from its air volume. It also allows for a broad comparison of FYI, SYI, and ridges. Meanwhile, we substantially shortened this section from 40 to 25 lines.

*Additionally, I suggest starting the section with the seasonality of air volume as the main topic of the manuscript (as explained in the following paragraphs starting from line 311) and only mentioning the other parameters, if needed for explanation.*

**Reply:** We agree with your suggestion and removed the subsection about "seasonal evolution of sea-ice temperature, salinity, freeboard, and snow depth".

*L313-314: I do not understand the citation of Golden et al. (1998) here, as it reads like reporting the observed seasonality in the data of the manuscript?*

**Reply:** We agree with your suggestion and clarified that the referenced paper defined the permeability threshold as 5% brine volume fraction.

*L335: "We focused on FYI physical properties…" is contradictory as before FYI properties are mentioned for different seasons.*

**Reply:** We removed the following reasoning of why we focused on FYI and described it as "less extensive sampling of SYI". We also shortened the subsection about SYI to 7 lines.

*L331-343: I am not able to follow the main message of this paragraph and its implications for the results before.*

**Reply:** We agree with your comments and shorten the discussion about SYI.

*L346-353: This paragraph seems to be misplaced and mixed of topics; instead of methods it includes comparison to another region (Antarctica) and seasonality. The next paragraph (starting from L354 contains a more fitting start of this subsection.)*

**Reply:** We agree with your suggestion and moved this paragraph, focused on intercomparison of our density estimates with historical observations, to a separate section "Comparison with previous measurements of sea-ice density," which was also substantially shortened to 7 lines. In addition, we added a reference to Fig. 1 presenting a historical overview of density measurements, including values presented in this study. We understand your concern about mentioning Antarctic measurements, but the referenced study covering this region is the rare one to present density in a seasonal context. We are not aware of similar overviews for Arctic sea ice.

*L354-368: This paragraph can be split, e.g., around L358, in half as the topic shifts from a comparison of the methods and highlighting hydrostatic weighing as the most accurate to the effect of brine loss.*

**Reply:** We agree with your suggestions and added a new paragraph.

*L369-390: This paragraph should be split around L376. Before, the effect of freeboard variability on the density is evaluated. The sentence with "Indeed" mentions standard deviation and links it later "this supports" (L378) to variability on different spatial scales, which is a new topic. The usage of "indeed" is also not appropriate here, because the two topics are not directly connected and if the second is supposed to be an explanation a different connection is needed.*

**Reply:** We agree with this argument and divided this discussion into separate paragraphs about effective density and weighing estimates, with relevant referenced studies.

*A second split would be beneficial around L385 with everything after "We demonstrated..." being a concise and well-written summary paragraph for this subsection.*

**Reply:** We agree with your suggestions and added a new paragraph.

*L404-410: The connection between the CryoSat-2 retrieval and why a more physics-based parametrization of density would help falls a bit short. It would be beneficial to mention that CryoSat-2 measures freeboards and needs density for the thickness measurements. Additionally, I do not understand why the shortcoming around moist snow (second to last sentence) leads to this conclusion in the last sentence.*

**Reply:** We agree with your comments and added a clarification that CryoSat-2 measures ice freeboard. We also wrote that while CryoSat-2 is expected to measure ice freeboard, the radar penetration of snow and SSL in summer is presumably low, which partially explains the estimates of CryoSat-2 ice thickness evolution mostly related to the snow and SSL melt, not to actual ice melt. This discussion is now placed in separate subsection 4.2.

*L412-434: This paragraph contains several controlling ideas and should be split. Until L417 the main idea is to establish why the air volume fraction is important. Afterwards the topics feel a bit mixed between how different seasons influence the air volume fraction (melt vs. growth) as well as if this is about the whole ice column or different areas (granular ice).*

**Reply:** We agree with your suggestion and added a new paragraph after line 418.

*L487-500: It is not fully clear why the detailed descriptions of the melt ponds are needed for the reader to follow the effect of their presences (drained or undrained) on the sea-ice density. I suggest shortening or removing this whole paragraph as the next paragraph directly addresses the topic of the subsection.*

**Reply:** We agree with your suggestion and removed most of the text from lines 487–500.

*L501-517: This paragraph was a bit hard to follow and could probably be improved by, e.g., grouping effects of undrained and drained melt ponds in one paragraph together. Additionally, at L512 the scale moves from local scale to larger scale.*

**Reply:** We agree with your suggestion and added a new paragraph starting at line 512.

*L525-541: I assume that this paragraph is including the effect of ridges on sea-ice density, but the topic of the paragraph changes from how ridges influence snow, to the effect of ridges on density via*

*a temperature effect (L535-536) and back to snow. I strongly suggest detangling these two effects in separate paragraphs.*

**Reply:** We agree with your argument. First, we moved the text about density measurements of ridges to a separate paragraph at the end of this section. Second, we added a better introduction to why this topic is discussed (line 525). Here we address how such a minor (on a first sight) and unrelated feature as an unrepresentative ridge fraction may lead to substantially larger snow thickness on a kilometer scale. This complicates the interpretation of pre-melt snow freeboard measurements outside of the area where snow thickness was directly measured. As we aim to include pre-melt estimates of sea-ice density as a reference, we also must acknowledge these limitations.

*L527-529: The goal stated in the sentence "To improve …" does not directly link to the theme of the manuscript being on the connection of air volume fraction on sea-ice density. I suggest therefore to remove this part and details on the found ridge fraction.*

**Reply:** We cannot agree with this statement. For the presented analysis of level ice density, it is important to separate ridges and undeformed ice. As we showed, ridges, mainly due to larger snow accumulation, have much larger snow freeboards, which should not be compared to the various measurements of level ice freeboard (Fig. 3b).

Therefore, if one wants to estimate ice freeboard from snow freeboard measurements, which are available on much larger scales than ice thickness or density, such classification should be performed. Otherwise, the existing measurements of snow thickness at level ice and ridges (Itkin et al.) cannot be properly used (without knowing level ice and ridge fractions at each scale). Level ice freeboard measurements are required to intercompare its summer evolution with direct measurements of level ice density.

As this study is focused on melt season, we also showed that ridges melt 3 times faster than level ice (Fig. 4d), making their direct freeboard comparison less valid for both winter and summer. We suggest that our classification allowed us to compare the summer evolution of level ice at the ice floe scale (1 km²), much larger than our draft measurements (0.3 km diameter). Summer evolution of level ice freeboard on larger scales similar to our small-scale measurements is directly related to the effect of sea-ice density evolution. And only air fraction increase allows for such freeboard increase during ice melt, while brine volume increase has an opposite effect decreasing ice freeboard.

*L534: The word "ridge density" is potentially confusing here, as before the paragraph talked about ridge fraction, but Figure 6g refers to sea-ice density.*

**Reply:** We agree with your suggestion and substituted "ridge density" to "sea-ice density of ridges" to avoid confusion.

*L576-577: I am unsure if this conclusion stems from the results of this manuscript – where in the results sections was internal melt or brine drain analyzed?*

**Reply:** Our interpretation of geochemical processes during ice melt is presented in Section 4.3. In brief, we showed how the ice surface experienced brine drainage and corresponding air enrichment without brine and air volume correlation. Meanwhile, below the waterline, internal melt in brine pockets and the corresponding density difference of ice to brine contributed to the air fraction increase, also supported by a correlation of air and brine volumes. We believe that interpretation of results should be presented in discussion, while we can also refer to the outcomes of such interpretation while presenting our conclusions.

***Reviewer RC3, Harry Heorton***

We thank the reviewer, Harry Heorton, for their helpful comments and the work they put into the review. Our answers are starting with "Reply" in bold, *following the original reviewers' comments in italics*.

*This paper documents to combination of an impressive volume of observational data from the MOSAiC Campaign. These are handled and documented impeccably allowing for them to be combined in order to estimate the density of the sea ice cover. The uncertainties in all the data along with the complications from several aspects of sea ice (ridges, melt ponds) are considered in order to present well documented and contextualized estimates of sea ice density. The work that is documented here is an important addition to sea-ice science and the completeness of the observational work may well make this paper the key text on any future work considering sea ice density. This paper is fit for publication with only some minor changes to the text.*

*The only suggestion that the authors may want to spend some time considering is the title. The current title doesn't cover the breadth of work covered here. The results cover both winter and melt seasons and the results are important for both knowledge of sea ice density and not just the effect it has toward the consideration of remotely sensed freeboard. A title such as "Impacts of air fraction increase on Arctic sea-ice density and freeboard" does the breadth and wider applicability of the results justice.*

**Reply:** We thank you for your evaluation of our work and suggestions on how to improve its title. We suggest a composite of two titles as "Impacts of air fraction increase on Arctic sea-ice density, freeboard, and thickness estimation during melt season." This combines the importance of air fraction for both "density and freeboard" but also for "thickness estimation." We suggest keeping "melt season" as one of the main outcomes of our study, which is that ice melt can be invisible for an altimeter without considering summer changes in air volume and ice density, while winter data is presented as a reference.

*Minor points as follows:*

*This is more of an editing issue than a review, but can all figures be made larger? Line width will probably be ok. A lot of zooming happened during this review.*

**Reply:** Unfortunately, we followed the journal guidelines also included in the LaTeX template. Figure width in TC should be 8.3 cm or 12 cm for one- or two-column figures. We fully agree with your recommendation and will address this issue with the editor. Sorry for the inconvenience. In the updated version, we will present figures with the full-page width.

*L 15 While this statement is true for observed mass balances shown in the next line – there is a growing body of work looking at the overall mass balance using thickness data, see Ricker et al. (2021). This sentence may confuse a reader with experience of these new works.*

**Reply:** We do not fully understand this comment. We claim that the goal of mass balance observations is to produce reliable ice thickness measurements throughout the year. We acknowledge the importance of the referenced work, but it used CryoSat-2 ice thickness estimates from Ricker et al. (2018) for the winter season only. Therefore, we do not think that our statement is incorrect.

*L 17 and onwards, this list is great, you may want to add very recent use of Passive Microwave too (Soriot et al. 2023)*

**Reply:** We cannot fully agree with your suggestion. In the referenced study, passive microwave measurements were used as a proxy for sea-ice thickness, not providing a direct observation. Considering that other referenced methods are providing direct observations and are widely used, we prefer not to add this reference.

*Early introduction - a precise definition of bulk density is needed. There are several subtly different measures of density included here and this makes the use of the term 'bulk density' difficult to follow. This is important to have for the comparisons that follow from L 57.*

**Reply:** We added a definition of ice bulk density in line 57 as "density of a composite material including pure ice, air, brine, and solid salts."

*L 83 – units needed for the salinity (and then throughout the paper in several places)*

**Reply:** We present all salinities on a practical salinity scale, which is dimensionless. This is a standard way to present salinity in oceanography. TEOS-10 recommends converting practical salinity to absolute salinity measured in g/kg, but this correction is not fully relevant for the Arctic Ocean, as its salt composition was not measured to give precise salinity anomaly values as it was done for other oceans. Units for practical salinity and difference between practical and absolute salinity can be found in McDougall et al. (2012, 10.5194/os-8-1123-2012). Since our salinity was measured using a conductivity meter, we present it as practical salinity as described in line 139.

*L 89 – some of these citations are not recent, does the dissolved state refer to one particular study here? This sentence can be re-written as it took a few reads to understand.*

**Reply:** We agree with your suggestion, but we also removed this part from introduction. We left a detailed explanation of this research in Section 4.5 of discussion.

*L 133, how was the direct freeboard and draft measured?*

**Reply:** We added a clarification that ice thickness and draft were measured using ice thickness gauge (plastic tape with a foldable metal bar at its end), with ice freeboard being a difference of those direct measurements.

*L 161 is the effective sea ice density a floe wide estimate using the whole floe freeboard?*

**Reply:** Not exactly. The effective ice density is different from ordinary density as it is estimated indirectly from the measurements of the ice freeboard and draft (as defined in line 161). It can equally include freeboard measurements for a single ice core or the whole ice floe. In this study, the effective ice density was estimated for coring (20-30 weekly point measurements of ice draft and freeboard) and for co-located laser scanner and underwater sonar surveys, limited by the size of ROV sonar surveys including first-year level ice (200 m in diameter). The effective ice density hasn't been estimated for the whole 1 sq. km ice floe as ice draft measurements are not available at this scale.

*Section 2.2, does the ALS freeboards assume the reflecting surface is the top of the snowpack? This needs to be precisely identified here.*

**Reply:** We believe that the very first sentence in line 181 tells that the Airborne Laser Scanner (ALS) measures "elevation of snow, snow-free ice, or melt pond." In addition, we clarified this in line 186 as "To calculate ice freeboard from snow freeboard measured by ALS." We hope that this makes it clear that ALS freeboard assumes the reflection from snow if snow/SSL are present.

*2.3 Linked to the previous point, in this section is the unponded ice freeboard at the sea ice to snow interface or the snow surface?*

**Reply:** We added additional clarification related to your question. Freeboard of ponded/unponded drained/undrained ice refers to ice freeboard, not snow freeboard. Snow thickness in Eq. (3-7) is only responsible for snow/SSL mass loading, present at unponded areas. To make it clearer, we added in lines 219, 225, and 230 that we provide equations for ice freeboard. We also wrote in line 236 that a snow freeboard can be found as a sum of ice freeboard fb_i and snow thickness h_sn.

*L 220 is hi ice thickness? I can't find it defined previously. It may be worth repeating here for clarity.*

**Reply:** The ice thickness notation was introduced a bit earlier, in line 164. But we also added the meaning of "hi" to line 222 as suggested.

*L 364 how does ice surface roughness affect this measurement?*

**Reply:** We have not found an elegant way to explain it as it mainly depends on how roughness is considered in the values of effective ice thickness and draft. Therefore, we removed this topic to make the paper clearer.

*L 385 Can this statement be re-written to show that this demonstration is an argument of the authors that can be made using the results of the study.*

**Reply:** This sentence is a summary of the whole Section 4.3. We added the following sentence: "The effective density estimates from the measured ice freeboard and draft of cores, and by ALS, ROV, and satellite observations, converge only after more than a hundred measurements are taken, reflecting uncertainties in other terms of the hydrostatic equation rather than true ice density variability."

*L 474 This - Thus*

**Reply:** We agree with the suggestion, it should be "thus" instead of "this". But we also had to delete this sentence completely to make the manuscript shorter following reviewers' suggestions.

*Paragraph 404-410. Does this rely on information within Landy et al 2022, or has data from this study been accessed to make this comparison? This may not need a full data section, but extra clarity here on how this information is created is needed.*

**Reply:** We agree that this should be better described. Here we reference a method paper of the gridded ice thickness product from Landy et al. we used to find ice thickness for MOSAiC location. We rewrote the mentioned sentence as "Sea-ice thickness estimate from ice freeboard measurements with CryoSat-2 radar altimeter by Landy and Dawson (2022) showed a strong ice melt of 1.5 m between late-May and June followed by 0.62 m melt in July." We hope that this makes the description clearer. We also moved this analysis to a separate Section 4.2.

*Figure 9, is it possible to adjust the solid blue circles in (b and d) to have outlines as it is very hard to see what size a lot of them are?*

**Reply:** We agree with your suggestion and added black outlines to blue circles in Fig. 9.

*L 544 'different densities' is this related to snow or sea-ice? Tricky sentence to follow*

**Reply:** We agree with your comment. Indeed, this was related to "different sea-ice densities." We had to delete this sentence completely to make the manuscript shorter following reviewers' suggestions.

*L 546 the link between surface roughness and snow thickness is not obvious. Is this a measurement effect of just the overall variability?*

**Reply:** The referenced study from Itkin et al. (2023) showed that ice roughness explained up to 85 % of the observed snow thickness variability, supporting our claim in line 416.

*L 548 onwards, are these measurements from this study or also from Itkin?*

**Reply:** We added a clarification that this (following) sentence presents our results, not findings from Itkin et al. (2023), which are presented in lines 545–548.

*L 569 'we use' – 'we present'*

**Reply:** We agree with your suggestion and made the corresponding changes.

*L 585 'the' melt season*

**Reply:** We agree with your suggestion and made the corresponding changes.

***Reviewer RC4***

We thank the reviewer for their helpful comments and the work they put into the review. Our answers are starting with "Reply" in bold, *following the original reviewers' comments in italics.*

*The manuscript aims to investigate how impacts of changes in internal ice properties (in particular, air fraction and brine volume, and its relation to sea ice density) translate into sea ice thickness retrievals, with a focus on the melt season. This is evaluated using an impressive and exhaustive number of data sources and methods from several different data sources acquired during the l Multidisciplinary drifting Observatory for the Study of the Arctic Climate (MOSAiC) expedition, including coring, in situ measurements (i.e. snow depth, freeboard, draft), thermistor-string buoys (ice mass balance/IMB buoys), upward looking sonar mounted on remote sensing vehicles (ROVs), and airborne observations of total freeboard (airborne laser scanner/ALS). During the melt season, they observed an unexpected increase in freeboard. In particular, the study focusses on changes in air fraction and brine volume to understand the driving factors of such phenomena which unaccounted for will have significant impacts of the derivation summer sea ice thickness retrievals derived from altimetry and the interpretation of such observations.*

*First, it is an impressive feat collecting this unique combination of data and combining them in such a matter, which makes it possible to evaluate all these distinct aspects – so, kudos to the authors for this impressive work! The methodology is sound, and overall, the results are well represented and well described.*

*However, I did have trouble reading the manuscript and following many of the conclusions – it took me several goes. Nonetheless, I believe this can be minimized with some reorganization of the sections and edits to the paragraphs for clarity. Overall, my suggested edits and comments are considered "major," which is not a reflection of the amount of work or its quality. It is however reflecting the significant re-organization I believe is necessary for a reader to fully understand and appreciate the detailed and thorough analysis. Furthermore, I urge the authors to consider some of the aspects of the airborne processing and their associated uncertainties when it comes to the observed increased freeboard changes, however I do not believe significant re-processing is required here.*

*Overall, with some reorganization of the manuscript, I strongly recommend publication of this work. It makes a relevant and critical contribution helpful in expanding our knowledge on sea ice summer processes and its impact on crucial sea ice geophysical variables derived from satellites.*

**Reply:** Dear reviewer, we thank you for your overall positive evaluation of our manuscript. We have done our best to address your main concerns, mainly by substantially reducing the manuscript length from 600 to 470 lines and reorganizing the sections and paragraphs. We have provided detailed replies to all your questions and suggestions below, attempting to provide enough referenced evidence.

*Major comments*

*Re-structuring and organizing the manuscript*

*As noted by all the other reviewers, the manuscript is long and hard to reach; the paragraphs are often long and include several mixed key take-aways and conclusions. While I support the other reviewers' overall comments regarding re-structuring, I here highlight some of the aspects that made it most difficult for me.*

**Reply:** We agree and removed some unnecessary information while also improving the manuscript's structure.

*Abstract: Currently reads more like a conclusion from the start, without a small introduction into the field or its importance. Consider 2-3 sentences on the hypothesis, importance etc.*

**Reply:** We agree with your comment, and we added two sentences to introduce the importance of sea-ice thickness observations and the current limitations.

*Results: The result section was exceptionally detailed, but difficult to follow. There are many different terms and techniques used, which are hard to remember when reading the manuscript.*

**Reply:** We agree with your comment, and we reduced the amount of information in the result section (from 110 to 70 lines) while compiling some of the results in Table 1.

*A way to refer to all the different variables and methods more easily, could be to use parameters instead of full text with proper, well-described sub-scripts? Often, this could also make it more difficult, but a table with a definition of the different variables, methods, and techniques along with the parameter in the beginning of the manuscript could aid readability. These parameters should then also be referenced in the figures.*

**Reply:** We agree with your criticism here. We think that the main issue with the way we initially presented our results is related to the complex comparison of density estimates from various methods. Therefore, we decided to have consistent names for these three methods and estimates from them, including (1) weighing density (2) coring density (3) density at the ROV site. We added this notation to our methods. Meanwhile, we would prefer not to use letters with subscripts, something like $\rho_{weighing}$, $\rho_{coring}$, and $\rho_{ROV}$, as we believe it would make reading more challenging.

*Please, consider highlighting some of your main numerical values in tables or in potentially in the figures. They get somewhat lost in the large body of text.*

**Reply:** We agree with your suggestion and added Table 1 with estimates of bulk density from all three methods together with the main physical ice properties, including salinity, temperature, air and brine volume, snow, and ice thickness. We also added a new Figure 1 with a comparison of historical overview and our observations of ice density measurements.

*Discussion: While this is one of the largest sections, it also has quite long paragraphs with multiple takeaways. Consider, if all aspects are truly relevant, to potentially make even more "sub-sections" and to make shorter paragraphs with one key take-away, to ease readability.*

**Reply:** We agree, and we added several additional sections and sub-sections, including "importance of density measurements" and "comparison with radar altimetry." We also significantly shortened the discussion, especially sections about melt ponds and ridges.

*Conclusions: While giving a great overview, I'm missing some of the larger-scale aspects where you results will have influence.*

**Reply:** We added the following concluding sentence to this section: «We showed that both our and historical observations of sea-ice density reveal its strong dependence on sea-ice temperature and salinity, with the range of summer density decrease potentially making around half of typical summer thickness loss not detectable from ice freeboard observations».

*Also, an aspect that I started considering reading the different reviews provided by the other reviewers was: who is the expected reader of this manuscript? I originally accepted to review this*

*manuscript, since I read that airborne altimetry observations would be used, and it had a focus on summer sea ice thickness derivation from altimetry – both topics I work with myself. However, reading the manuscript, many distinct aspects came into view, and I was unsure who this manuscript was really targeted. Also, from the different reviews, it seems that different people took aspects of the manuscript as a main focus. E.g., sometimes, it was noticeably clear that the focus related to "altimetry-derived thickness estimates," other times it was more related to the ice properties and in situ observations of such. Granted, I know that it is all inter-connected, but I do urge you, during this re-structuring, to consider the aim of your manuscript and who you are targeting. This will likely also help you streamline your result and discussion section a bit more.*

**Reply:** We agree with your suggestion, but we think that there is no simple reply to this question about the potential audience. Yes, it aims to introduce seasonality of sea-ice density in ice thickness retrieval algorithms for remote sensing observations. Meanwhile, it also provides an overview of how we can measure sea ice density to obtain accurate results. In addition, it provides only a simple parametrization of ice density vs. its temperature, while a more complex model should also be introduced to describe air volume evolution in sea ice. We believe that a synergy of accurate ice observations, providing simple parametrization for remote sensing, and validation values for geochemical modeling should be considered in future studies. We think that the disconnect between those separate fields led to a current gap of knowledge, which may be fixed only using multidisciplinary research. Meanwhile, we significantly reduced the length of sections not directly related to either the thickness estimate from the snow/ice freeboard or the air fraction effect on ice density. This mainly includes parts of introduction and discussion related to ridges, melt ponds, and historical overview of density measurements.

*Interpretation of airborne (ALS) observations and their importance*

*Now, from how I read the manuscript, one of the main drivers for this study related to an unexpected increase of sea ice freeboard during the melt season – driven to the assumption, that ice is melting which overall, we would expect a decrease in thickness (and, intuitively, also in the freeboard). However, since altimetric thickness-derived measurements rely on the assumption of hydrostatic equilibrium, the buoyancy, and the changing internal properties of the ice (driven by the summer processes) appear to counter-act this, complicating the process and to some extent, invalidating the assumptions applied to the altimetry observations. And, since I primarily work with remote sensing altimetry observations, I will keep my main focus on this aspect for technical considerations.*

*I do worry somewhat about the certainty of which the ALS observations are being presented. The average freeboard increases are stated to be 1-3 centimeters! That is hardly within the accuracy of the airborne observations themselves (which you state is 2.5 cm), and surely not within the uncertainty of the ALS observations of freeboards (an elevation uncertainty of 5 cm is stated in your data section). I would have liked some discussion on uncertainty estimates of the freeboard values, which are provided in the ALS data products, especially related to how the freeboard results compare within those uncertainties.*

**Reply:** First, we agree with your statement that we cannot provide an estimate of freeboard change below ALS uncertainty. Therefore, for each separate ALS scan, the estimate is within the mentioned range of 2–3 cm. The main message of our study is that ice freeboard evolution based on ALS shows an increase of 1–3 cm, while ice melted by 0.6 m and its freeboard should have been decreased by 6 cm instead assuming constant ice density. This difference between 1–3 cm increase and 6 cm decrease should be compared with ALS uncertainty instead.

Second, following your question, we estimated the standard error for ALS full scan surveys during the winter season during December–May. As previously shown by Koo et al. (2021, 10.1016/j.rse.2021.112730), this period was characterized by a linear increase in ice and snow thickness both for small scales (buoys) and large scales (estimates from IceSat-2 snow freeboard measurements). We calculated both mean and modal snow freeboard for all 17 ALS scans performed during December–May. The standard error was 2.1 cm and 2.9 cm for ALS modal (representing level ice) and mean freeboards. This agrees well with elevation uncertainties of 2.5 cm given in Hutter et al. (2023). Assuming the same errors are applicable to melt season (characterized by large areas of open water that are used for ALS freeboard correction), we suggest that the density effect on freeboard evolution of around 8 cm is substantially (3 times) larger than ALS uncertainties of 2.5 cm. We added this estimate to line 127.

[Figure]

Figure: Modal (left) and mean (right) freeboard for full ALS surveys during the freezing season.

In addition, the freeboard change between late June and late July is nearly identical on various scales, from 200 m by 200 m of ROV site to 1 km² of CO2 ice flow and 40 km² of full ALS coverage. Similarly, during spring when both snow and ice thicknesses were relatively unchanging, ALS showed nearly identical snow freeboard for four scans in March-May.

We added the confidence interval to the attached figures. Meanwhile, we do not think that we should present ALS modal and mean freeboards during the winter and spring seasons, as it is not relevant to the focus of our study. We attached the corresponding figure to this response.

*A straightforward way to showcase the uncertainties – or spatial variability the average freeboard estimates– could be with a confidence interval in your plots. Now, I do recognize that an increase of a similar magnitude was also observed by the in-situ estimates at coring sites (with significantly lower uncertainty and better accuracy), but for the larger scale surveys, this is relevant considering the spatial variability and the different processing/data that goes into your sea ice freeboard estimation. Especially, in relation to the impact of snow (which you do mention and discuss too!).*

**Reply:** We agree with your suggestion and mentioned the confidence intervals of ALS modal freeboard, representing level ice according to Koo et al., 2021. We also agree that coring sites provide accurate estimates of snow or ice freeboard only after considering several coring events, which agrees with our findings as well as from Hutchings et al. (2005).

We should mention that the spatial variability of snow is not relevant for the melt season. In July, the spatial variability of SSL was significantly lower than that of snow, and SSL thickness was also not sensitive to ice type in contrast to snow.

*ALS methodology*

*The ALS observations are, as you state, snow freeboard (or total freeboard) – or, in the absence of snow, the (sea) ice freeboard (or the surface scattering layer, I believe you also define it as)? So, for spring observations, there is a need to remove the snow estimates. Now, this you have done by considering near-daily estimates of Magnaprobe observations (using either average CO2 transect data on ALS full scale, or level ice average transect data at ROV site). I am curious about your considerations for this, especially in terms of the spatial variability. You state that the snow cover is quite heterogenous – so, how come one average value be representative of these large scales? Do the studies (e.g., Itkin et al., 2023) state that there was little spatial variability over the FYI level ice site (if so, please report it to support this choice). And, for the CO2/full survey site – is there not a better representation that could be made from the snow estimates? How were the transects performed across CO2, and would there be any benefit it better representing this spatial variability (in particular over the rougher ice/near ridges), rather than using the average data – and if so, could this be implemented? Or do you expect a small impact on this for the spring estimates?*

**Reply:** We agree with your suggestions and arguments. We added a brief intercomparison of level FYI snow thickness from IMBs, coring sites, and transect to Section 4.4 (line 311–315), which shows that the average values were nearly identical.

For the first question, Itkin et al. showed that (1) the smallest snow thickness variability was for level ice and (2) on the scale of CO1 and CO2 snow thickness converged for more and less deformed transects. We also added our estimates of snow thickness from two coring sites and two buoys located within 100 m from the ROV site. Their estimates of snow thickness were nearly identical between each other and similar to transect estimates for level ice and IMB estimates for FYI. Considering that FYI ROV site covered similar ice area to linear transect observations, there is a good confidence in its representativeness for CO and CO2 level ice.

For the second question, we do not think that there is a better snow thickness dataset for CO2 ice floe than from transect measurements, which covered a substantial part of the floe edge (Itkin et al., 2023). Meanwhile, we added to the discussion that transect snow thickness should not be extended beyond CO2 ice floe as it may give large errors in ice freeboard and density estimates. We justified it by a non-representative fraction of ridges (our work) and SYI (Kortum et al., 2024) for CO2 in comparison to surroundings (full ALS coverage and beyond).

The last question was about methodology and snow thickness for different ice types. We used average transect snow thickness for the whole CO2 and level ice transect snow thickness for level ice areas of CO2, including FYI ROV site.

We fully agree that such observations cannot be simply upscaled without consideration of surface roughness, which is responsible for 85% of snow thickness variability (Itkin et al., 2023). Therefore, as we already mentioned, usage of transect snow thickness for areas outside of sampled ice floes (CO1, CO2) may lead to high errors in ice freeboard estimates. Which leads to the conclusion that our spring ice freeboard estimates for the full ALS coverage are uncertain. And such methodology may give substantial errors without an overlap with in-situ measurements as for our ROV and CO2 sites. We also clarified that such methodology is not recommended for autumn-spring periods unless snow thickness is measured in parallel, as in Jutila et al. (2022). These limitations are not applicable for the summer season, while the airborne method remains the most uncertain among the presented.

*You state that ALS does not provide freeboard of melt-ponds directly below the helicopter, from what I expect, is an impact of specular scattering. How do you define what is melt ponds (and should be*

*(bi)linearly (?) interpolated from the edges), and what is in fact open water? This was not clear from the text.*

**Reply:** We added the following sentence to our methods: "Gaps in freeboard for the ALS data over ponds at the nadir of the helicopter survey were filled by bilinearly interpolating freeboard at the pond edges." First, we referenced the data publication from Hutter et al. (2023), where the method of retrieving freeboard from elevation and open water detection is described in detail. This statement is an outcome of that publication, not the current study. In our study, we mostly focused on small patches of ice without any open water (ROV site and CO2 ice floe). We, of course, double-checked if the freeboard of nearby open water is indeed zero, but this was not a part of the analysis, and the published freeboard dataset was not reprocessed or modified. For the specific melt ponds, we intercompared areas with no freeboard measurements and orthomosaics RGB images to double-check if those areas are indeed ponded. Together with ALS snow/SSL/melt pond freeboard measurements, we have co-located measurements of ice draft, photogrammetric melt pond depth and freeboard, and aerial images (Fig. 1b). We think that any of these data may show that our sampling above the ROV site was performed over ponded ice and not open water.

Minor comments

Title

Currently, the title does not reflect the full set of results and discussions presented in the manuscript and might suffer from this mixed presentation of impact of the internal properties and its relation to remote sensing techniques. I would suggest you reconsider the title, in the frame of considering your expected reader and which results are the main results you want to present.

**Reply:** We changed the title to "Impacts of air fraction increase on Arctic sea-ice density, freeboard, and thickness estimation during melt season." We do not think that title may or should cover the whole scope of the paper. We agree that a significant part of our study is about comparing density estimates using various methods, but this is done mainly to validate rapid changes in density due to air volume increase upon warming.

*Figures*

*While the figures are well made, and nicely present the results, I must agree with the other reviewers, that they were hard to read due to the small size. I also had to zoom in several times, and in the printed version, many of the conclusions were not possible to derive from the figures. I suggest you increase the figure size to text width, which might now follow the TC template (which, I believe, states widths of 8 cm). However, I would overall recommend you increase the size so that the figure label font size correlates more or less with the size of the text in figure captions.*

**Reply:** We agree with your suggestions and changed the figure width from 12 cm, suggested in TC template, to text width. We are sorry for the inconvenience, but we were unsure how TC preprint validation would handle such modification in the template. We will ask TC editors about this issue.

*Specific suggestions to improve readability and easy understanding of these (overall) great, but also complex figures:*

*Figure 1:*

*It is not entirely clear for me how sub-panel c was generated.*

*There were 6 coring events, right, represented by the numbers? In that case, why is there a "gap" in collection, if this does not represent "continuous" measurements, say from IMBs, but from cores? And how are the contours in-between cores generated?*

**Reply:** Thank you for your questions and suggestions. We added clarification at the caption that black circles represent coring events. Second, the gap represents a break in the x-axis representing time, as with a linear scale, 1 May is much further from 20 June than in the figure, which would make readability harder. Third, yes, you are right; as for a typical contour plot, it presents linearly interpolated values between measurements. We assumed that this is the standard way to present any discrete time series.

*What does the "blue" shadowed area represent – the freeboard? Or the snow?*

**Reply:** We mentioned in Fig. 1 caption that gray-shaded areas represent snow or surface scattering layer. The ice freeboard is shown as a black solid line.

Could you highlight the "zero"-line (if that is presenting the water line)? The "increase in freeboard," which I believe you are also presenting here, is not very clear with the contour overlaid.

**Reply:** That is correct; zero values of the y-axis representing depth in meters are equivalent to a waterline. We agree with your suggestion and add a black dashed line representing waterline to make ice freeboard evolution more apparent.

Also, it is not noted what the circles in sub-panel (a) represent.

**Reply:** We agree with your suggestion and added a clarification that black solid lines represent ice surface and bottom interfaces with black round markers representing each coring event.

Figure 3: Not all the information is easily deduced from the plot.

Sub-panel a: What is "snow ROV" and "ice ROV"? I suspect it is related to the ROV and coring sites, since the ROV itself cannot separate snow and ice freeboards? But how are these freeboards measured/derived?

**Reply:** We agree with your suggestion and added a clarification to the figure caption about methods by which freeboard, draft, density, and volume fractions were measured and estimated for the two presented sites, FYI coring and FYI ROV sites. We also substituted "ROV" and "coring" to "ROV site" and "coring site" in legends to make the reference to sites clearer.

Sub-panel b-c: What goes into the definition of "level" here? Consider including this in the caption.

**Reply:** We agree with your suggestion and added a clarification to the figure caption that "level" refers to ice outside of ridged areas defined by classification from Section 2.2.

*Sub-panel e: The last line of the legend is not easily readable.*

**Reply:** We increased the width of the red dotted line to improve its visibility.

*Figure 4:*

*Consider including an additional column with a difference plot of both freeboard and draft; that is, the overall difference (or trend) from the 10th of May to 22nd of July (or show an initial pre-melt example and then differences to that for each subsequent sub-panel). While overall changes are easily observable, the locations of most pronounced changes are lost.*

**Reply:** We attached the figure with freeboard change relative to pre-melt values on 10 May to this response. We have not found a good way of presenting this data to contribute to our study. The temporal evolution of sea-ice draft during melt season was analyzed in the other study (10.5194/tc-17-4873-2023). Similarly, spatial meter-scale variability of ice melt is outside of the scope of the current manuscript.

There are several reasons why this haven't been shown in this specific figure as follows, (1) we show measured snow/SSL/melt pond freeboard, as we cannot estimate ice freeboard for each location of ALS scan due to no information about snow thickness distribution, (2) difference in spring and summer freeboard would give an impression of the difference between thicker snow depth and thinner SSL depth for unponded areas and melt pond freeboard for ponded areas, (3) while snow thickness distribution on level ice might be interesting, it is outside of this study's focus, (4) meter scale co-location of ice freeboard might be imperfect and lead to the corresponding artefacts, while in this study we mainly focused on the average values for the selected FYI ROV site.

A more comprehensive analysis of ice draft evolution in summer (10.5194/tc-17-4873-2023) showed a strong positive correlation of ice draft and melt, with enhanced ridge melt. We think that figures with locations of larger draft changes would only redirect a reader towards a different complex topic of which ice melts faster. And while ice thickness change affects hydrostatic balance, this is out of the scope of this study. In addition, bottom melt is only a small fraction of the total snow and ice melt, as was shown by Smith et al. (10.5194/egusphere-2024-1977). Meanwhile, our data might not be a perfect source for analysis of surface melt rates as (1) our co-location is worse than for direct measurements (2) we cannot distinguish freeboard changes due to snow and ice melt. To keep our study focused on summer evolution of ice density, we decided not to present this analysis.

[Figure]

Figure: Snow/SSL freeboard change between 10 May and 30 June (a), 17 July (b), and 22 July (c).

*Also, the colormap here confuses with its diverging colors. If you do not show differences, I would suggest a sequential colormap.*

**Reply:** We agree with your suggestion, and we changed the colormap from diverging "broc" to sequential "batlow" from Crameri et al. (2020) in Fig. 4.

*Also, I would be interested in similar freeboard/difference in freeboard maps from the CO2 and larger scale surveys. Would that be possible to include, to understand the spatial variability?*

**Reply:** We agree that it might be interesting, and we are happy to provide the data and the scripts (which are publicly available). But such analysis would be outside of the scope of the study. The spatial variability of the freeboard change can be evaluated based on the figure we attached to one of

the previous questions. In brief, the standard deviation is slightly larger than the mean value and is increasing during the melt season. The most extreme values are associated with areas with larger surface roughness. As many suggestions from other reviewers were towards making the study more focused, we think that such additions may have an opposite effect.

*Figure 8. How come the plots and values here look different than Figure 4? Aren't the sub-panel a+d repeats? Still not sure about the diverging colormaps, I also suggest a sequential here unless you are trying to highlight some difference (e.g., in the density by low/high density contrast).*

**Reply:** The only difference between these two figures is that here we show snow/SSL freeboard with filled gaps at a fraction of melt ponds not scanned by ALS, while in Fig. 4 we showed raw ALS data without this modification. This is done to have a cleaner picture of the effective ice density in panels (b) and (e). We modified Fig. 4 to match Fig. 8. Similarly to Fig. 8, we changed the colormap to sequential "batlow."

*Abbreviations*

*Check that all abbreviations and acronyms are defined. For example, MOSAiC is not defined until the data availability section but should be defined in both the abstract and first time used in the main text.*

**Reply:** We agree with your suggestion and added MOSAiC acronym meaning to line 112 of introduction. We believe that other acronyms (FYI, SYI, MYI, ROV, ALS) are defined in our manuscript.

*Data availability section*

*I appreciate that all the data is publicly available. However, I would urge the authors to also consider providing the data processing and plotting scripts (e.g., via a GitHub repository) in the name of Open Research.*

**Reply:** We agree with your suggestion. The script for data processing and plotting is added to GitHub: https://github.com/esalganik/density.

*Technical corrections*

*While I recognize that a large reconstruction of the manuscript is likely to also the paragraphs and the text written, hence some technical corrections may render irrelevant, I still present a few that I noticed while reading.*

*The definition of surface scattering layer "SSL" is not that clearly presented in the manuscript, and it can be hard to distinguish in the discussion/results why you use this term at times, and why snow at other times. Consider, when you define this term, to include a succinct explanation of why you make this separation.*

**Reply:** We cannot fully agree with this statement. Surface Scattering Layer (SSL) was defined in line 162 as "deteriorated granular melting ice similar to large-grained melting snow" with reference to a study about this sub-material by Macfarlane et al. (2023). Following the definition, we now added clarification that "Due to its granular structure, snow and SSL could not be distinguished during coring and transect measurements, and here we refer to both snow and SSL as snow." More details about snow and SSL formation and evolution can be found in Webster et al. (2022), while Macfarlane et al. (2023) provide a more detailed overview of SSL formation and evolution. In our study, SSL was treated as snow due to the nature of Magnaprobe and coring sampling, which cannot distinguish these two materials. Similarly, from the laser altimeter point of view, SSL is similar to snow. We tried

to be consistent and mention that transect and snow pit measurement provide observations of both snow and SSL. In rare cases, snow was used instead of snow/SSL for simplicity and readability. We checked all mentions of snow in our manuscript and added "/SSL" when we discuss melt season observations.

*Line 2. "we observed the first-year (FYI) freeboard increase by "to "we observed a first-year (FYI) freeboard increase of"*

**Reply:** We agree with your suggestion and made the corresponding changes.

*Line 8. "co-located ice topography from" to "co-located ice topography observations from"*

**Reply:** We agree with your suggestion and made the corresponding changes.

*Line 12. "from 0.92 to 0.87 observed FYI" to "from 0.92 to 0.87 observed over FYI"*

**Reply:** We agree with your suggestion, but for consistency, we substituted this text with a concluding sentence about air volume and density seasonal evolution.

*Line 13. "from satellite altimeters during" to "from satellite altimeters under assumption of hydrostatic equilibrium during"*

**Reply:** We agree that this might be a more accurate description, but we also think that the referenced sentence is already too long for providing this level of detail in an abstract. After all, this is a generally accepted method. Instead, we added this clarification to line 21 of Introduction as "To convert draft or freeboard to sea-ice thickness under assumption of hydrostatic equilibrium, remote methods measuring freeboard or draft require information on the snow depth and density of both snow and sea ice."

*Line 17. "laser altimeter (ICESat) and radar (Sentinel-3)" to "laser altimeter (ICESat, ICESat-2) and radar (e.g., Sentinel-3, CryoSat-2)"*

**Reply:** We agree with your suggestion and made the corresponding changes.

*Line 21. In "(height above the waterline)" and "(height below the waterline)," consider using elevation instead of heights. Not sure we really use "heights below" something?*

**Reply:** We agree with your suggestion and made the corresponding changes.

*Line 26. Not sure "require" is the right word here? "observe" perhaps?*

**Reply:** We agree with your suggestion, but this sentence was removed to improve readability of Introduction.

*Line 28-49: Perhaps consider mentioning that Alexandrov et al. MYI density was based on an estimated value from upper- and lower-layer ice density estimates.*

**Reply:** We agree with your suggestion, and we removed the MYI density estimate from Alexandrov et al. (2010) due to both potentially inaccurate assumptions behind it and to improve readability and focus on first-year ice observations.

*Line 53. What is meant by "performed at the ice in situ temperatures"? Seems like a word might be missing.*

**Reply:** This refers to temperatures at which ice density is measured. Usually, sea-ice cores are cooled down to a low laboratory temperature, at which their density is eventually measured. Measuring

density at low laboratory temperatures means that brine volume is much lower than at in situ conditions. We clarified that we refer to "density measurements" and reformulated this part as "in situ temperatures of ice."

Line 55. I would suggest a reference to Jutila et al. 2022.

**Reply:** We agree with your suggestion and added the corresponding reference.

Line 57-69. Why a sudden intro to Antarctic sea ice too? This is not really investigated or truly discussed further in the entire manuscript, as far as I read. Could potentially be removed for clarity and size reduction.

**Reply:** We cannot fully agree with this statement. First, unlike for Arctic observations, the study from Fons et al. (2023) provides a seasonality of Antarctic first-year ice density. To our knowledge, such seasonal density parametrization was not published for Arctic sea ice. We added a historical overview of Arctic in situ measurements of FYI density to Fig. 1 to make this more apparent (there are nearly no measurements of sea-ice density for summer and autumn). We also compared our seasonal observations of FYI in Section 4.2 of Discussion. We think that it is worth mentioning that this seasonality is quite similar for both Polar regions.

Line 82. Is matrix the right word here?

**Reply:** We removed this sentence to improve the readability of the Introduction. Meanwhile, the word matrix is often used in this context, e.g., in Golden (2001, 10.3189/172756401781818329).

Line 83. No "unit" on the salinity?

**Reply:** According to TEOS-10, practical salinity (measured by conductivity meters) is dimensionless, and this is the standard way to present practical salinity unless it is converted to absolute salinity measured in g/kg as in Schulz et al. (2024, 10.1525/elementa.2023.00114). Since in the referenced study the practical salinity was reported without this conversion, we present it dimensionless (like our measurements).

*Line 88-89. "Given that small changes (…)" – this seems to be the primary premise for this work! However, this is not well reflected in the findings.*

**Reply:** We agree and substituted this sentence with "Given that changes in the air fraction have a greater impact on sea-ice density than changes in brine or ice volume".

*Line 296-297. "Previous studies (…)" – this statement could do with a reference.*

**Reply:** We agree with this suggestion and added references to Cottier et al. (1999) and Notz (2005).

Sub-section headline, line 180. Missing abbreviations of ALS and ROV?

**Reply:** We presented abbreviation of ALS in line 181 and of ROV in line 193. We followed your suggestions and added them to the section's title.

*Line 198. Could you include a sentence on how false bottoms are detected, and therefore, possible to distinguish?*

**Reply:** We agree with your suggestion and added a clarification that "false bottoms were detected using temporal evolution of ice draft measured by sonar before and after the local melt pond drainage." In the referenced study, we identified the condition of false bottom formation defined by the presence of ridges around the area with most false bottoms. The ice area surveyed in this study

was located close to a floe edge, which allowed under-ice meltwater to migrate upwards into open water areas, which led to no detectable false bottoms.

*Level 209-210. Quite certain that Ricker et al. (2023) uses 0.6 m above modal elevation as threshold for winter ridge detections.*

**Reply:** We agree that Ricker et al. (2023) used a 0.6 m threshold for ridge detection (i.e., ridge or not ridge). Meanwhile, here we refer to 0.5 m used in the same study to evaluate sail areal fraction "fraction of elevations >0.5 m (5.3 % for gt2r)," as in our methodology we also aimed to evaluate area of sails. Please correct us if we got it wrong. We substituted "following" for "similar to" to avoid potential misinterpretation. But, in both cases, our ridge keel estimates are not sensitive to the chosen threshold as we used co-located ridge surface and bottom topography.

*Line 256. What observations that deviated from other values? How do you consider them as deviating? And how many observations was this?*

**Reply:** Here we refer to a single ALS survey, which showed much lower snow freeboard of the investigated areas (ROV site and the whole CO2 ice floe) than all other ALS surveys before and after. The same survey was also excluded from the melt pond airborne photogrammetry study by Fuchs et al. (2024). These errors were possibly caused by the misidentification of melt ponds as open water during the conversion from elevation to freeboard.

*Line 271. I do not know what SSL density means? You reference Macfarlane et al. (2023) here, but this sentence is not clear for me. Why relevant?*

**Reply:** SSL density is the density of the surface scattering layer. This material's properties, together with snow density, were presented in Macfarlane et al. (2023). It is relevant to our study as during most of July ice was covered with a thin (5–10 cm) layer of very porous SSL with a low density close to 400 kg/m³, which was not considered ice by coring and Magnaprobe observations. The density of SSL was used to calculate snow loading for hydrostatic balance calculations in our study, as defined in the mentioned Section 3.1.

*Line 350. How did FYI show better agreement than SYI (either reference to a Table or provide a measure of how far they deviate)? – otherwise, with all the numbers, it can be hard to identify which from your results to compare with.*

**Reply:** We agree with your suggestion, and we added references to the new overview Table 1 and overview Fig. 1, while also adding values from our study for an easier comparison.

*Line 352: I do not understand how the airborne pre-melt standard deviation is 4 kg m⁻³, when you state later that airborne densities usually range around almost 30 kg m⁻³. This value seems exceptionally low for an airborne estimate.*

**Reply:** We agree that the meaning of the presented standard deviation could be explained better. The current standard deviation only represents variability between average density estimates for the four ALS freeboard surveys in March–May. It does not represent the standard deviation of each point estimate in contrast to Fig. 8. This was partially mentioned in Fig. 5 caption, saying that «error bars represent one standard deviation of weekly measurements in summer or of all pre-melt weekly measurements». We added a clarification to line 352 as "standard deviation is given for average estimates from four ALS scans in March–May." This was made for a fair comparison with other more precise methods, as standard deviations for freeboard/draft estimates from coring also represent variability of weekly average densities, not single drillholes. We devoted Section 4.4 to the

comparison of different methods and showing larger uncertainties for ALS, especially if local snow depth is not measured as in Jutila et al. (2023).

*Line 404-410. While great to include this aspect of satellite-derived summer sea ice thickness from CryoSat-2, I am having a challenging time following it and the conclusions.*

**Reply:** We agree with your evaluation and substantially reformulated this paragraph, which is now converted into a separate section 4.2 and illustrated in Fig. 7c. We attempted to show that during both snowmelt in May-June and ice melt in July, CryoSat-2 was not able to penetrate snow and SSL, resulting in its freeboard representing snow freeboard. Therefore, CryoSat-2 ice melt estimates (assuming constant ice density) of 1.5 m in May-June and 0.6 m in July were identical to ALS-derived snow freeboard decreases of 0.14 and 0.05 m. In our study, we showed that on different scales, the ice freeboard was nearly unchanging during ice melt in July. This suggests that both ice density and SSL thickness evolution should be considered to improve CryoSat-2's ability to obtain ice melt instead of snow/SSL melt.

*Please, at least, provide a link to the data/manuscript when first mentioning this data, since you do not have it in the data section.*

**Reply:** We agree with your suggestion and added both references to the CryoSat-2 ice thickness algorithm from Landy et al. (2022) and to the corresponding dataset by Landy and Dawson (2022) in the data availability section.

*I would have loved to see a figure on how the data looked around the site! You state that a similar decrease is observed surrounding the CO2 site (at 80 km resolution). Please, consider providing a map example of this data including the data around this site supporting this info. Could be included e.g., in Fig 1 as an additional sub-panel.*

**Reply:** We added an estimate of accumulated ice melt from CryoSat-2 ice thickness estimates from Landy and Dawson (2022) to Fig. 7d. Landy and Dawson (2022) provide a year-round gridded ice thickness estimate with 80 km resolution. This estimate is for the closest to MOSAiC ice floe grid point. Meanwhile, we are not sure that providing a map with thickness estimates may be illustrative. This is because of the large error in CryoSat-2 estimates, probably related to snow penetration. Therefore, showing apparently biased ice thickness or ice melt values might not be helpful for our study. We attached a comparison of ice thickness and freeboard from IMB, ALS, and CryoSat-2 to this response. Since for most of the melt season the CryoSat-2 estimates are not fitting the observations, it is hard to select a date to show estimates for the surrounding area.

*Why are you mentioning May-June 2022? And IMBs compared with, are they also from this time then? This seems like a sentence, where the ramifications/impact from this observed difference is not fully explained to the reader. Why include it? Perhaps expand on it in case this is relevant.*

**Reply:** We are sorry about the wrong time mentioned in line 406; it should be, of course, 2020, not 2022. We separated these two sentences and added the context. In brief, we used ice melt estimates from IMBs because (1) they covered a larger area than CO2 comparable to CryoSat-2 grid size in Landy et al. (2022) and (2) because there were no other MOSAiC ice mass balance measurements except for remotely working IMBs in May and most of June. We added an intercomparison of CryoSat-2 and IMB thickness evolution to make this clearer.

*I am not sure what is meant by the statements in Line 408-410. Why mentioning of moist snow/SSL to CryoSat-2 – why does this relate to the ALS decrease of 0.14 m and 0.05 m?*

**Reply:** We agree with your arguments and substantially rewrote this paragraph to explain why we think CryoSat-2 measures snow/SSL freeboard during melt season.

*The last sentence, which I believe must refer to this impact during the melt season, is interesting, but needs to be expanded upon further. What is meant by SSL thickness – how to even consider this? Or, have this been considered before?*

**Reply:** We defined what a surface scattering layer (SSL) is in line 162. SSL thickness stands for the thickness of this layer. In line 187, we mentioned that we mainly used SSL thickness measurements from Magnaprobe transects, as Magnaprobe can measure the thickness of any granular material, such as snow and SSL, but cannot distinguish them from each other. Transect measurements, including Magnaprobe snow and SSL thickness, are presented in Webster et al. (2022, 10.1525/elementa.2021.000072); here they are shown in Fig. 6a. SSL thickness was also measured manually using thickness tape, as was done at the coring site. In our paper, similarly to Webster et al., we treat SSL as snow. Therefore, SSL thickness was considered in all measurements that required getting an ice freeboard from snow/SSL freeboard, measured by laser scanner (ALS). SSL was also considered in hydrostatic balance similar to snow mass load, as its density was measured using a density cutter, not ice coring, due to the granular structure of SSL. We also provided a detailed description of snow and SSL thickness evolution in lines 266-271. We think that we described SSL definition and measurements with enough details, also providing references to papers that explored various aspects of this layer in more detail.

As SSL is visually indistinguishable from snow, following Webster et al. (2022), we considered it as snow, not sea ice. During summer, radar altimeters might not be able to penetrate both snow and SSL. Therefore, we attempted to compare CryoSat-2 ice thickness estimates with the evolution of both SSL and ice freeboards, showing that this lack of penetration agrees with altimeter estimates. We suggest that both evolution of sea-ice density and SSL thickness play an important role in interpretation of CryoSat-2 freeboard measurements in summer. Whether one should classify SSL as snow or sea ice is beyond this study's scope.

*Reviewer CC1, Arttu Jutila*

We thank Arttu Jutila for their helpful comments. Our answers are starting with "Reply" in bold following the original reviewers' comments in italic.

*Dear Evgenii and co-authors,*

*Congratulations on your hard work on the challenging topic of sea-ice density! You have studied a very important and impactful topic, which is clearly reflected in the number and encouraging content of the already posted referee comments.*

**Reply:** Dear Arttu, thank you for the positive feedback and thank you for the inspiration given by you pioneering study from 2022.

*With this community comment, I would like to discuss and clarify a statement related to L351-352 of your manuscript: "Unlike our observations, which showed similar FYI and SYI thickness (Fig. 6b), Jutila et al. (2022) observed SYI to be 3.2–4.5 m thick, 3–6 times thicker than adjacent FYI."*

*Sea-ice thickness results of different ice types in the analyzed IceBird campaigns were not discussed in Jutila et al. (2022). I believe you must have extracted the IceBird SYI thickness range directly from the 2019 data set (Jutila et al., 2024). However, the data does not appear in your reference list nor in the data availability section. In the name of good scientific practice and journal data policy, I believe the reference should be included.*

**Reply:** We agree with your statement, and we added the updated version of the dataset referenced in your 2022 manuscript to the reference list. Thank you for checking our text and your useful feedback.

*That said, I have no doubt about the correctness of the SYI thickness range. However, I think it is important to understand the underlying sea-ice type definitions. According to the definition applied in Jutila et al. (2022), sea-ice type was determined using a combination of sea-ice thickness data and NSIDC's weekly 12.5-km sea-ice age data product. In short, the thickness threshold was chosen such that FYI and SYI cannot have the same thickness, i.e., FYI had a thickness of less than 2 m (upper limit dictated by thermodynamic growth and WMO's Sea Ice Nomenclature) whereas SYI had a thickness of at least 2 m. Also deformed ice was not excluded. More details can be found in Section 2.5.2 Sea-ice type of Jutila et al. (2022).*

*I recognize the fact that the sea-ice type classification scheme in Jutila et al. (2022) is far from perfect but perhaps adequate considering the available data and spatial scales reaching to regional transects several hundred kilometers long.*

**Reply:** Thank you for the clarification. We updated the reference of your study to make the used classification visible. As you might have seen, this sentence was used for (1) showing limitation of our SYI density estimates as densities of underformed relatively thin ice (2) explaining differences between your and our SYI densities despite better agreement in ALS-derived FYI densities. Of course, for our observations covering different scales, we could not use identical ice classification.

And many thanks for the recent paper led by Karl Kortum with your co-authorship (10.5194/tc-18-2207-2024), we found it well-made and very useful, especially for providing fractions of different ice classes including FYI, SYI and deformed ice.

---

## Referee Report (RR1)

I would like to thank the authors for implementing the reviewer suggestions. Overall, the revised manuscript has improved significantly. The authors have greatly improved the readability of the manuscript. The purpose of the work is now clearly articulated. The result and discussion sections are concise and easy to read. Well done.

Suggested corrections:

- In-situ vs in situ: The authors use both forms of the word. Consider choosing either the hyphenated form or the unhyphenated form for consistency.
- Figure 2 Caption: "Panel (c) shows the contour plot of FYI density evolution from hydrostatic weighing with bulk density values for each coring event shown in blue and grey shaded areas representing snow or surface scattering layer thickness." Consider placing a comma "(…) shown in blue, and grey shaded areas (….)".
- Table 1 Caption: "Bulk physical properties of level first-year ice at the coring and ROV sites" A period is missing at the end of the caption.
- Figure 4 caption: "Dotted blue line in (b) represents sea-ice density estimate corrected for coring measurements performed in unponded areas as described in Section 4.6." Consider adding an article at the beginning of the sentence, "The dotted blue line in (b) …"
- Figure 4 caption (and Figure 2 caption): "Time axis is not continuous." Consider adding an article at the beginning of the sentence, "The time axis is not continuous."
- Figure 7 panel (d): The scatter plot shows FYI in blue and SYI in orange, while the lines show FYI in orange and SYI in blue. The legend labels or line colors might have been mixed up here.
- Figure 10: I think something went wrong when rendering the figure. The legends are not aligned. Lines delimiting the individual legend boxes are missing. Additional boxes are floating left of panel (b) and panel (e).
- Lines 471-472: "Upscaling of these estimates during winter is complicated by the large-scale spatial variability of snow depth due to variability of ridge fraction." Consider changing the sentence to "(…) is complicated due to the large-scale variability of snow depth and/given the variability of the ridge fraction."

---

## Author Response (AR2)

***Reviewer RC1, Alice Petry***

We thank the reviewer, Alice Petry, for their helpful comments and the work they put into the review. Our answers start with "Reply" in bold, *following the original reviewers' comments in italics*.

*General comments:*

*I would like to thank the authors for implementing the reviewer suggestions. Overall, the revised manuscript has improved significantly. The authors have greatly improved the readability of the manuscript. The purpose of the work is now clearly articulated. The result and discussion sections are concise and easy to read. Well done.*

**Reply:** Dear Alice, we thank you for your positive evaluation of our manuscript and your detailed reviews which helped us to improve the readability of our paper.

*Minor comments / suggested corrections:*

*In-situ vs in situ: The authors use both forms of the word. Consider choosing either the hyphenated form or the unhyphenated form for consistency.*

**Reply:** We thank you for the suggestion and changed all the versions of "in situ" to its unhyphenated form, as it follows the journal guidelines.

*Figure 2 Caption: "Panel (c) shows the contour plot of FYI density evolution from hydrostatic weighing with bulk density values for each coring event shown in blue and grey shaded areas representing snow or surface scattering layer thickness." Consider placing a comma "(…) shown in blue, and grey shaded areas (….)".*

**Reply:** We thank you for the suggestion and added the suggested comma to the caption.

*Table 1 Caption: "Bulk physical properties of level first-year ice at the coring and ROV sites" A period is missing at the end of the caption.*

**Reply:** We thank you for the suggestion and added the suggested period to the caption.

*Figure 4 caption: "Dotted blue line in (b) represents sea-ice density estimate corrected for coring measurements performed in unponded areas as described in Section 4.6." Consider adding an article at the beginning of the sentence, "The dotted blue line in (b) …".*

**Reply:** We thank you for the suggestion and added the suggested article to the caption.

*Figure 4 caption (and Figure 2 caption): "Time axis is not continuous." Consider adding an article at the beginning of the sentence, "The time axis is not continuous."*

**Reply:** We thank you for the suggestion and added the suggested article to the two captions.

*- Figure 7 panel (d): The scatter plot shows FYI in blue and SYI in orange, while the lines show FYI in orange and SYI in blue. The legend labels or line colours might have been mixed up here.*

**Reply:** We thank you for noticing this error. We swapped colors of lines representing FYI and SYI temperatures to be consistent with other panels.

*Figure 10: I think something went wrong when rendering the figure. The legends are not aligned. Lines delimiting the individual legend boxes are missing. Additional boxes are floating left of panel (b) and panel (e).*

**Reply:** We thank you for the suggestion and aligned horizontal lines of panels and legend in Fig. 10. The legend is shared between different panels, so delimiting lines were not added.

*Lines 471-472: "Upscaling of these estimates during winter is complicated by the large-scale spatial variability of snow depth due to variability of ridge fraction." Consider changing the sentence to "(...) is complicated due to the large-scale variability of snow depth and/given the variability of the ridge fraction."*

**Reply:** We thank you for the suggestion and adjusted the text as you recommended.

*Reviewer RC2:*

We thank the reviewer for their helpful comments and the work they put into the review. Our answers are starting with "Reply" in bold, following *the original reviewers' comments in italics*.

*I want to thank the authors for the work they put into the revision of the manuscript. The manuscript improved a lot, and the scientific contribution presented very well. I only have a couple of minor comments, which mostly are technical.*

**Reply:** Dear reviewer, we thank you for your positive and constructive evaluation of our manuscript.

*Minor comments:*

*L35: The word "accurate" is confusing here, maybe it can be removed. Why does the data needs to be extended if it is already accurate?*

**Reply:** We agree with your suggestion and removed the word "accurate".

*L47 and L56: "outside of the melt season" - the article "the" is missing.*

**Reply:** We agree with your suggestion and added "the" to "outside of melt season".

*L115: There is an article missing before coring site. Is it "the coring side" or "each coring side".*

**Reply:** We agree with your suggestion and added "each" in front of "coring site".

*Figure 7: "ridge" is sometimes written with a capital R or a small r in the legend. Maybe the figure would become less busy, if it had one shared legend outside of the plots similar to Figure 10.*

**Reply:** We agree with your suggestion and capitalized "ridge" in Fig. 7g. Unfortunately, legends within eight panels of Fig. 7 are mostly different, with five various contents. This makes a shared legend not very practical.

*L276: I suggest to add a descriptor/noun after "this" to enhance reading comprehension, e.g., "this discrepancy". There are also other places where "this" is alone with a descriptor (e.g., L281 and L322) and could benefit from it.*

**Reply:** We agree with your suggestion and added "this discrepancy", "this potential explanation", and "these observations" instead of "this" in lines 276, 281, and 322, respectively.

*L278: "may be seen" could be replaced with either "could be seen" or even a stronger implication.*

**Reply:** We agree with your suggestion and substituted "may" with "could".

*L290-291: It is not clear to me, how a different "sea-ice density parametrization" helps alleviating an effect of snow load? Maybe this could be explained with another additional sentence.*

**Reply:** We agree with your comment and added the following clarification "This is due to the observed snow freeboard decrease in July being equivalent to the CryoSat-2 ice thickness decrease (Fig 5b)". This relates to the observed snow freeboard decrease and ice freeboard increase during melt season in July. Using constant ice density would not allow for any ice melt unless snow freeboard is not false identified as ice freeboard due to low radar penetration.

*L295: From your reviewer comments I understand that the purpose of the Antarctica densities is to compare seasonality. Nevertheless, this is for me not clear in this section of the text yet. Additionally, it is not clear what values afterwards are for the Arctic and the Antarctic. Maybe this needs to be two*

*different paragraphs, as the ideas are a) comparison to other seasonal products and b) comparison to other Arctic values or at least somehow made clear what values are for which region.*

**Reply:** We agree with your comment and added a more general introduction about a similarity between temperature dependence of our and historical observations for Arctic sea ice. We also moved a comparison between Arctic and Antarctic densities to the end of this paragraph.

*L319: The units display density, while you talk about freeboard. Is there something mixed up?*

**Reply:** We are sorry for this mistake, we substituted "freeboard" to "sea-ice density" in that sentence.

*Figure 10: There is no dashed line in e). Additionally, I am not sure if I understand what the dashed lines shows. Is it supposed to show the air volume increase based on a density change?*

**Reply:** We thank you for the suggestion and added a dashed line illustrating 10% of brine volume to Figure 10e. Yes, as described in the caption, a line showing air volume equal to 10% of brine volume represents the potential air volume evolution upon warming due to ice and water density differences. This assumption is satisfied if air bubbles are at constant pressure, unlike for our FYI observations.

*L319: The units display density, while you talk about freeboard. Is there something mixed up?*

**Reply:** We thank you for noticing this error. Indeed, the presented values are standard deviations of density, not freeboard. We changed "freeboard" to "sea-ice density" in this sentence.

*L348: Do you have observed that the ice is columnar in the ice cores or is this an assumption, that the ice is columnar because of its region and age? It would be great if you could clarify that and if possible mention, if/where the ice in the measured profiles in Figure 8 switches from granular into columnar.*

**Reply:** We used the structure characterization based on thin sections of MOSAiC ice cores. One of them is presented in Nicolaus et al. (2022). We hope that all of them will be published by the corresponding principal investigators.

*L387: "typical for FYI" instead of "of"*

**Reply:** We thank you for the suggestion and adjusted the text as you recommended.

***Reviewer RC4***

We thank the reviewer for their helpful comments and the work they put into the review. Our answers are starting with "Reply" in bold, *following the original reviewers' comments in italics*.

*First, thank you to the authors for the great responses to four reviews and a community comment! Considering so many comments, which are also reflected in your response document (33 pages!), it must have been quite some work!*

*I am very happy with the responses to my comments and the edits made by the authors overall. I believe this version is ready for publication! I did find some (very) minor technical corrections, some of which I am sure will also just be caught in copy-editing by TC. Nonetheless, I list them here for you to keep an eye on. Great work and thank you for this interesting work!*

**Reply:** Dear reviewer, we thank you for your overall positive evaluation of our manuscript.

Minor comments / technical corrections

Line 44. Consider removing "mentioned earlier" and simply referencing Jutila et al. 2022 after "various combinations of remote methods" for clarity.

**Reply:** We thank you for the suggestion and removed "mentioned earlier" from that sentence.

*Line 83. Remove start parenthesis before 2023b in "(ALS, Hutter et al. (2023b)."*

**Reply:** We agree with your suggestion and removed the parenthesis.

*Section 2.3. I might have missed it, but I do not see a reference to Figure 3, even though it seems to be a schematic explaining these different melt pond drained/undrained parametrisations.*

**Reply:** Thank you for noticing. We added a reference to Fig. 3 at line 166, where we introduce surface topography classification.

*Figure 7. Just for your information, the legend for subfigure (g) is written "ridge," but (h) is "Ridge."*

**Reply:** We thank you for the suggestion and made the legend's caption capitalized.

Line 263. Remove start parenthesis before 2010 in "(e.g., Alexandrov et al. (2010)".

**Reply:** We agree with your suggestion and removed the parenthesis.

Line 278. Is "may" the correct word here? Is it, or is it not seen in your observations?

**Reply:** We agree with your comment. Indeed, we can definitely say that our observations showed the absence of ice freeboard increase during snowmelt in May and June.

Line 431. Suggest rewording "above" to "around". I am not convinced that much snow accumulates on top of the ridges but rather around them due to wind redistribution.

**Reply:** We think that the study by Itkin et al. (2023) showed that ridges also accumulated a lot of snow above them. Average snow thickness in April was 0.6 m and 0.14 m for ridges and level ice, respectively. In addition, they estimated that snow drift extended around 25 m to each side of the sail crest. The total length of this snow accumulation of 50 m is comparable to a typical ridge keel width of 30 m (e.g., Ekeberg et al., doi:10.1016/j.coldregions.2014.10.003). Therefore, we changed "above" to "above and around". We hope that this is a fair compromise supported by the field measurements.

"Data availability" and "Code and data availability" sections. Please include the "Data availability" text in the "Code and data availability" section now that both are relevant.

**Reply:** We thank you for noticing this error. We moved reference to the code to "Code and data availability" section with data references.

*Additional changes*

Here we list additional minor changes we implemented in addition to the ones in response to reviewers' comments:

- In several instances, "snow depth" was substituted for "snow thickness" for consistency.
- In Fig. 1, we changed the legend text from "Alexandrov" to "Alexandrov et al."
- In Fig. 2a we added a color bar with bathymetric depth.
- In Fig 6, we added to the caption an explanation of the "sn," "fb," and "d" values displayed in the figure.
- We added to section 4.3 text describing similarities of our sea-ice density observations with the parametrization suggested in Fons et al. (2023) for Antarctic sea ice, suggesting that their parametrization can be also used for Arctic sea ice.
- In the "code and data availability" section, we substituted a GitHub URL for a Zenodo publication based on the same repository with updates following this revision.
- In line 155, we added a reference to a study describing physical properties of surface scattering layer, as this material is often discussed in our manuscript.